# The acyl- and alkyl-glycerone phosphate reductase DHRS7 is involved in the production of distinct plasmalogen species from DHRS7B

Tenga Takahashi[1], Kento Otsuka[1], Takayuki Sassa[1,2,*] and Akio Kihara[1,*]

## ABSTRACT

Plasmalogens are ether lipids that constitute mammalian membranes. In their biosynthetic pathway, DHRS7B functions as an alkylglycerone phosphate reductase. Here, we have investigated whether DHRS7, which shares high sequence similarity with DHRS7B, also contributes to plasmalogen production. We generated *DHRS7* and *DHRS7B* knockout (KO) cells, as well as double-KO cells, and quantified ethanolamine plasmalogens (PE[P]s). DHRS7 and DHRS7B were involved in the production of distinct fatty acid species: in the *de novo* synthesis pathway, DHRS7 contributed most strongly to the synthesis of C18:1 species, followed by C16:0 and C22:6 species, whereas DHRS7B contributed to the synthesis of all species examined. These differences in contribution in the *de novo* pathway resulted in further differences in the steady-state composition of PE[P]s. DHRS7 and DHRS7B also showed different intracellular localization: endoplasmic reticulum for DHRS7 and peroxisomes for DHRS7B. *In vitro*, DHRS7 exhibited both alkyl- and acyl-glycerone phosphate reductase activities, similar to DHRS7B. Collectively, our findings indicate that DHRS7 is a previously unreported acyl- and alkyl-glycerone (acyl/alkyl-glycerone) phosphate reductase that differs from DHRS7B in both subcellular localization and the profile of plasmalogen species produced.

KEY WORDS: Phospholipid, Ether lipid, Plasmalogen, Lipid synthesis, Mass spectrometry, Alkylglycerone phosphate reductase

## INTRODUCTION

Ether-linked glycerolipids contain a fatty alcohol linked at the *sn*-1 position of the glycerol backbone. They include ether-linked glycerophospholipids (e.g. plasmalogens and platelet-activating factors), glycolipids (e.g. seminolipids and glycosylphosphatidylinositol anchors) and non-polar glycerolipids such as ether-linked triacylglycerols (Dorninger et al., 2020; Schooneveldt et al., 2022). Plasmalogens are widely distributed across various tissues as constituents of cellular membranes and account for ~20% of total phospholipids in humans (Braverman and

[1]Faculty of Pharmaceutical Sciences, Hokkaido University, Sapporo 060-0812, Japan. [2]Department of Life Science, Kyushu Sangyo University, Fukuoka 813-8503, Japan.

*Authors for correspondence (sassat@ip.kyusan-u.ac.jp; kihara@pharm.hokudai.ac.jp)

T.T., 0009-0002-1939-4006; K.O., 0000-0002-6542-7414; T.S., 0000-0003-3145-9829; A.K., 0000-0001-5889-0788

Moser, 2012; Nagan and Zoeller, 2001). They are characterized by a vinyl ether bond at the *sn*-1 position and, as major lipid components of myelin, they are especially important for maintaining neural function, among various other physiological roles (Aggarwal et al., 2011; Dorninger et al., 2020). Seminolipids are specifically present in spermatozoa and spermatogenic cells within the testis, where they play an essential role in sperm formation (Honke et al., 2002; Tadano-Aritomi et al., 2003; Tamazawa et al., 2025; Tanphaichitr et al., 2018). Platelet-activating factors are lipid mediators and are involved in immune responses, such as the activation of mast cells (Tremblay et al., 2022). Glycosylphosphatidylinositol anchors, a form of post-translational protein modification, mediate the attachment of proteins to the outer leaflet of the plasma membrane (Paulick and Bertozzi, 2008). Thus, ether-linked glycerolipids are important lipids with functions that cannot be replaced by ester-linked glycerolipids.

The first two reactions in ether-linked glycerolipid synthesis occur in peroxisomes (Fig. 1) (Dorninger et al., 2022; Hajra, 1995; Nagan and Zoeller, 2001). In the first reaction, glycerone phosphate (GnP, also known as dihydroxyacetone phosphate), produced via glycolysis or gluconeogenesis, is acylated by GnP *O*-acyltransferase to produce acyl-GnPs (Thai et al., 1997). Acyl-GnPs are then converted to alkyl-GnPs by replacing their acyl moiety with a fatty alcohol, a reaction catalyzed by alkylglycerone phosphate synthase (AGPS) (de Vet et al., 1997). The fatty alcohols used in this reaction are predominantly produced from acyl-CoAs by the fatty acyl-CoA reductase FAR1, with a minor contribution from FAR2 (Takahashi et al., 2025). The GnP moiety of alkyl-GnPs is converted to a glycerol-3-phosphate (G3P) by the acyl- and alkyl-GnP (acyl/alkyl-GnP) reductase DHRS7B (dehydrogenase/reductase 7B), which is localized primarily in peroxisomes and partly in the endoplasmic reticulum (ER), resulting in the synthesis of 1-alkyl-G3Ps [1-alkyl lysophosphatidic acids (LPAs); 1-alkyl-LPAs] (Honsho et al., 2020; Lodhi et al., 2017). Subsequently, 1-alkyl-LPAs undergo acylation or, in the case of platelet-activating factors, acetylation at the *sn*-2 position, followed by the formation of a phosphodiester-linked head group at the *sn*-3 position. In the case of the biosynthesis of ethanolamine plasmalogens (PE[P]s), also known as plasmenylethanolamines, ethanolamine is first added to form plasmanylethanolamines (PE[O]s), which are then converted to PE[P]s by the plasmanylethanolamine desaturase TMEM189 through desaturation of the carbon–carbon bond adjacent to the ether bond, forming the vinyl–ether linkage (Dorninger et al., 2022; Gallego-García et al., 2019; Hajra, 1995; Wainberg et al., 2021; Werner et al., 2020). The acyl/alkyl-GnP reductase DHRS7B also catalyzes the conversion of acyl-GnPs into 1-acyl-G3Ps (LPAs) (Honsho et al., 2020; James et al., 1997), which are further metabolized via phosphatidic acids (PAs) to ester-linked glycerolipids.

DHRS7B is a member of the short-chain dehydrogenase/reductase (SDR) family and contains a class IV NADPH-binding motif in its

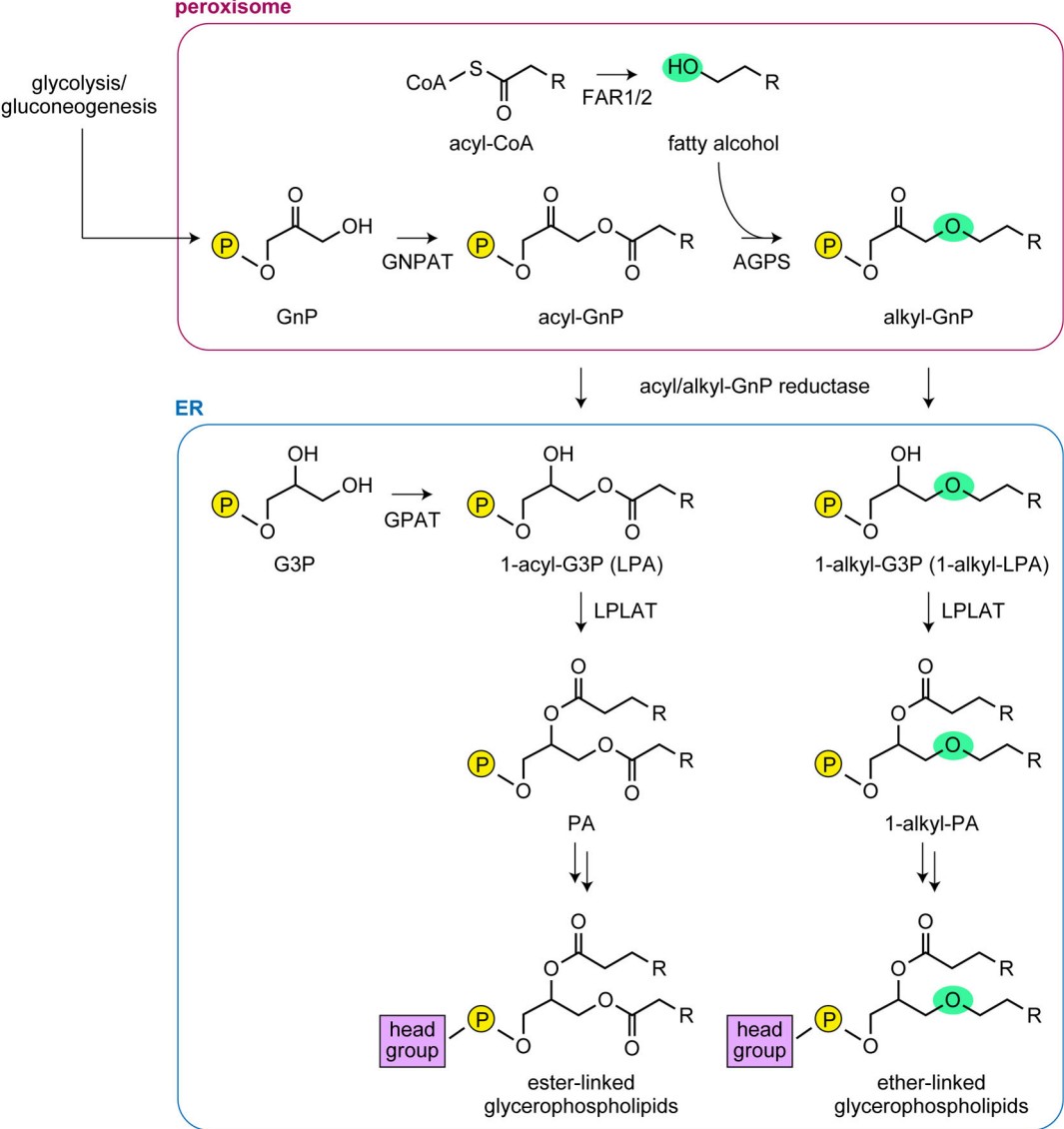

**Fig. 1. Biosynthetic pathway of ether-linked glycerophospholipids.** Ether-linked glycerophospholipid synthesis begins in peroxisomes with the acylation of GnP by GnP *O*-acyltransferase (GNPAT) to form acyl-GnP. AGPS then replaces the acyl group with an ether-linked fatty alcohol, mainly supplied by FAR1 (with a minor contribution from FAR2), producing alkyl-GnP. Alkyl-GnP is subsequently reduced at its carbonyl group by (an) acyl/alkyl-GnP reductase(s) to form 1-alkyl-G3P (1-alkyl-LPA). In the endoplasmic reticulum (ER), 1-alkyl-LPA is converted to 1-alkyl-PA by lysophospholipid acyltransferases (LPLATs), followed by further modifications to produce ether-linked glycerophospholipids. Alternatively, acyl-GnP can be reduced to 1-acyl-G3P by (an) acyl/alkyl-GnP reductase(s) for ester-linked glycerophospholipid synthesis. GPAT, G3P acyltransferase.

N-terminal region (Fig. S1A). The identification of *Dhrs7b* originated from the isolation of FAA.K1B cells, a mutant clone of CHO-K1 cells that was unable to incorporate exogenous fatty alcohol ([³H]hexadecanol) into complex lipids (James et al., 1997; James and Zoeller, 1997). FAA.K1B cells exhibited a reduction in [³H]hexadecanol incorporation to ~60% of control levels, while their acyl/alkyl-GnP reductase activity was markedly decreased, to about 5% of the control. Subsequent investigation revealed that FAA.K1B cells harbor a missense mutation (G194D) in the NADPH-binding motif of *Dhrs7b* (Honsho et al., 2020). In *Dhrs7b* knockout (KO) mice, choline-containing phospholipids in gonadal white adipose tissue were analyzed (Lodhi et al., 2017). In that study, only one ether-linked glycerophospholipid species, plasmanylcholine (PC[O]), was detected and found to be reduced relative to controls, although the analysis was not quantitative. In tamoxifen-inducible *Dhrs7b*

conditional KO mice, four PC[O] species were detected in bone marrow leukocytes, and they were reduced to 25–50% of control levels (Lodhi et al., 2015). Collectively, these findings indicate that DHRS7B plays an important role as an acyl/alkyl-GnP reductase in ether-linked glycerophospholipid biosynthesis. However, they also suggest that the existence of another acyl/alkyl-GnP reductase with overlapping function or an alternative biosynthetic pathway that bypasses the reduction of alkyl-GnPs.

DHRS7B belongs to the DHRS7 subfamily of the SDR family. This subfamily also includes DHRS7 and DHRS7C (Fig. S1B), which share 35% and 45% amino acid sequence identity with DHRS7B, respectively. Because this subfamily occupies a phylogenetically close position to enzymes with steroid-metabolizing functions (e.g. HSD11B1 and HSD17B4) and retinoid-metabolizing functions (e.g. RDH10), DHRS7 and DHRS7C have been investigated as potential

steroid- and retinoid-metabolizing enzymes. *In vitro* experiments showed that DHRS7 exhibited reductase activity producing cortisol, testosterone and all-*trans*-retinol (Araya et al., 2017; Štambergová et al., 2016). In experiments where DHRS7C was overexpressed in cells, incubation of the cells with all-*trans*-retinol resulted in an increase in all-*trans*-retinal (Treves et al., 2011). These *in vitro* and overexpression studies suggest that DHRS7 and DHRS7C can catalyze reactions related to steroid and retinoid metabolism; however, their physiological roles *in vivo* remain unclear, and it is possible that their primary function is as acyl/alkyl-GnP reductases in ether-linked glycerolipid biosynthesis. Therefore, in this study, we investigated whether these proteins could function redundantly with DHRS7B as acyl/alkyl-GnP reductases.

## RESULTS

### Tissue distribution of *Dhrs7* subfamily members

To investigate the tissue distribution of *Dhrs7* subfamily members, we examined their mRNA levels in 17 mouse tissues via quantitative real-time RT-PCR. Both *Dhrs7* and *Dhrs7b* were expressed in a wide range of tissues (Fig. 2). *Dhrs7* showed its highest expression in the eyelid, followed by the submandibular gland and prostate. *Dhrs7b* was most abundantly expressed in skeletal muscle, with the small intestine and white adipose tissue showing the next highest levels. In contrast, *Dhrs7c* expression was largely confined to the striated muscles of skeletal muscle and heart, with the highest expression in skeletal muscle, while low but measurable levels were also observed in the tongue. Among the *Dhrs7* subfamily genes, *Dhrs7c* exhibited the highest expression in both skeletal muscle and heart. Tissues in which *Dhrs7* showed predominant expression within the subfamily included the eyelid, submandibular gland, prostate, tongue and lung, whereas *Dhrs7b* was the most highly expressed subfamily member in the liver, kidney, small intestine, colon, brain and thymus. In white adipose tissue, eyeball, stomach and spleen, *Dhrs7* and *Dhrs7b* were expressed at comparable levels. These results reveal distinct tissue-specific expression profiles of the three *Dhrs7* subfamily members, implying that each may make different contributions to ether-linked glycerolipid biosynthesis in a tissue-dependent manner.

### Generation of *DHRS7* subfamily KO cells

To evaluate the role of *DHRS7* subfamily members in plasmalogen synthesis, we generated *DHRS7* KO, *DHRS7B* KO, *DHRS7/7B* double KO (DKO) and *DHRS7/7B/7C* triple KO (TKO) cells,

using the leukemia-derived, near-haploid cell line HAP1 and the CRISPR-Cas9 system. Guide RNAs were designed to target exon 4 of *DHRS7*, exon 3 of *DHRS7B* and exon 2 of *DHRS7C* (Fig. 3A). Two independent clones were established for the single KO cells of *DHRS7* and *DHRS7B*, whereas one clone was obtained for the DKO and TKO cells (Fig. 3B). Examination of the expression of each *DHRS7* subfamily member in HAP1 cells via quantitative real-time RT-PCR revealed that *DHRS7* and *DHRS7B* were expressed, while *DHRS7C* was not detected (Fig. 3C). We therefore primarily used the single KO cells of *DHRS7* and *DHRS7B*, and the DKO cells in the following experiments, and only included TKO cells in some of the experiments.

### Involvement of *DHRS7* in plasmalogen synthesis

To examine the effect of the gene KO of *DHRS7* subfamily members on lipid composition, lipids prepared from control, *DHRS7* KO, *DHRS7B* KO, *DHRS7/DHRS7B* DKO and *DHRS7/DHRS7B/DHRS7C* TKO cells were separated via thin-layer chromatography (TLC) and visualized via cupric acetate/phosphoric acid staining. No clear differences in the quantities of phosphatidylethanolamines (PEs)/PE[P]s, phosphatidylinositols/phosphatidylserines, phosphatidylcholines or sphingomyelins were detected among these clones (Fig. 4A; the slashes in these terms indicate lipid classes that co-migrate under TLC conditions and thus were not resolved). To distinguish PEs from PE[P]s and obtain more quantitative data, we next performed liquid chromatography–tandem mass spectrometry (LC–MS/MS) analysis. The total quantities of PE[P]s in *DHRS7* KO and *DHRS7B* KO cells (two independent clones for each KO) were reduced to about 60% of those in control cells, whereas *DHRS7/DHRS7B* DKO cells showed a more pronounced reduction to 34% of control cells (Fig. 4B).

PE[P]s contained alkyl chains of C16:0, C18:0 or C18:1 at the *sn-1* position and acyl chains of C16:0, C18:1, C20:4, C22:4 or C22:6 at the *sn-2* position, with the *sn-2* acyl-chain abundance following the order C18:1>C20:4>C22:6>C22:4>C16:0 (Fig. 4C). Hereafter, alkyl chains at the *sn-1* position are denoted with the prefix 'P-' to indicate the vinyl ether linkage characteristic of plasmalogens, whereas acyl chains at the *sn-2* position are shown without this prefix (e.g. P-C16:0 and C16:0). By convention, the double bond of the vinyl ether linkage at the *sn-1* position is not counted toward the unsaturation number; therefore, 'P-C18:1' corresponds to an 18-carbon chain with one double bond, plus the vinyl ether double bond. In *DHRS7* KO cells, the levels of C18:1

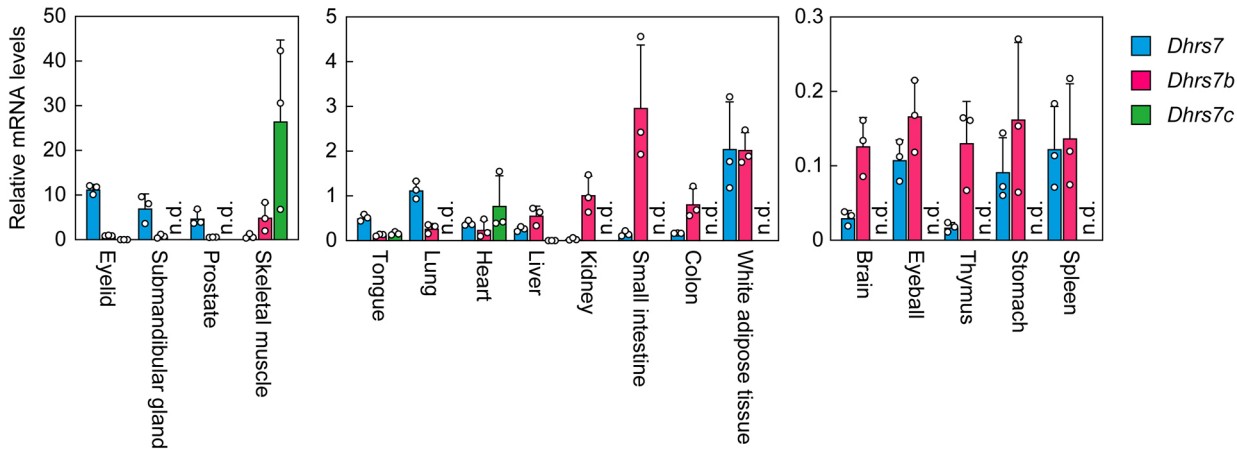

**Fig. 2. Tissue distribution of the *Dhrs7* subfamily in mice.** Total RNAs were extracted from the indicated tissues of 8-week-old male C57BL/6 mice and subjected to quantitative real-time RT-PCR using specific primers for *Dhrs7*, *Dhrs7b*, *Dhrs7c* or the housekeeping gene *Hprt1*. Data are mean+s.d. (*n*=3) of each mRNA quantity relative to that of *Hprt1*. n.d., not detected.

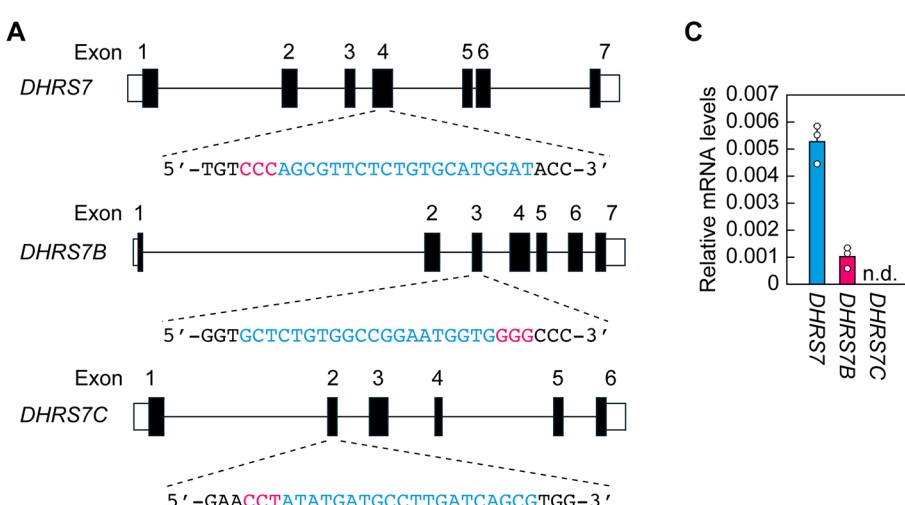

**Fig. 3. Generation of *DHRS7* subfamily KO cells.** (A) The gene structures of *DHRS7*, *DHRS7B* and *DHRS7C* (coding regions shown in black and untranslated regions in white) are shown, along with the nucleotide sequences of wild-type HAP1 cells around the guide RNA sequences (blue) and the protospacer-adjacent motif sequences (red). (B) The genotypes and mutation types of the *DHRS7*, *DHRS7B* and *DHRS7C* alleles in single, double and triple KO HAP1 cell lines generated via the CRISPR–Cas9 system are shown. (C) Total RNAs were extracted from HAP1 cells and subjected to quantitative real-time RT-PCR using specific primers for *DHRS7*, *DHRS7B*, *DHRS7C* or the housekeeping gene *GAPDH*. Data are mean+s.d. (*n*=3) of each mRNA quantity relative to that of *GAPDH*. n.d., not detected.

| HAP1 clones | *DHRS7* alleles | *DHRS7B* alleles | *DHRS7C* alleles |
|---|---|---|---|
| *DHRS7* KO1 | 1 bp insertion | WT | WT |
| *DHRS7* KO2 | 1 bp insertion | WT | WT |
| *DHRS7B* KO1 | WT | 14 bp deletion | WT |
| *DHRS7B* KO2 | WT | 2 bp deletion | WT |
| *DHRS7/7B* DKO | 14 bp deletion | 14 bp deletion | WT |
| *DHRS7/7B/7C* TKO | 2 bp deletion | 1 bp deletion | 38 bp deletion |

fatty acid (FA) species of PE[P] (including P-C16:0/C18:1, P-C18:0/C18:1 and P-C18:1/C18:1) were markedly reduced compared to control cells, regardless of the alkyl chains, with an average reduction to 25% of control levels. Species with C22:6 showed a moderate reduction (56%), while those with C20:4 were mostly unchanged relative to controls. In contrast, species with C16:0 or C22:4 showed an increase. In *DHRS7B* KO cells, all PE[P] species were reduced (Fig. 4C). Among these, species containing C22:4 showed a marked decrease, to an average of 42% of control levels, whereas species with other FAs exhibited more moderate reductions (50–60% of control levels). In DKO cells, most changes in individual FA species were not the result of a simple additive effect of the respective single KOs, but rather showed a more complex pattern. In DKO cells, C16:0 species were nearly absent, C18:1 species were reduced to *DHRS7* KO levels, C20:4 and C22:4 species were reduced to *DHRS7B* KO levels, and C22:6 species declined by roughly the sum of the reductions seen in both single KOs. In contrast to the PE[P]s, the total quantities of PEs in *DHRS7/DHRS7B* DKO cells were similar to those in control cells, even though they were slightly reduced in *DHRS7* KO and *DHRS7B* KO cells (Fig. 4D). The modest reduction in PE seen in the single KOs may reflect indirect influences, such as compensatory changes in gene expression or alterations in lipid homeostasis.

In the above experiments, we observed the steady-state composition of cellular PE[P]s. FAs in plasmalogens are initially introduced through the *de novo* synthesis pathway and subsequently undergo remodeling (Ford and Gross, 1994; Yamashita et al., 1997). Therefore, the steady-state composition reflects contributions from both the *de novo* synthesis and remodeling pathways, making interpretation complex. To specifically assess the FA composition introduced via the *de novo* synthesis pathway, we next performed a labeling experiment using $d_{35}$-C18:0 FA (stearic acid containing 35 deuterium atoms). Cells were labeled with $d_{35}$-C18:0 FA for 3 h, and $d_{35}$-labeled PE[P]s and PEs were quantified via LC–MS/MS. The $d_{35}$-C18:0 FA taken up by cells is converted to

$d_{35}$-C18:0-CoA, some of which then undergo chain elongation and desaturation. The resulting acyl-CoAs are either utilized for ester-linked lipid synthesis, including PEs, or, after conversion into fatty alcohols by the fatty acyl-CoA reductase FAR1 (Takahashi et al., 2025), used for synthesis of ether-linked lipids, including PE[P]s.

The FA composition of $d_{35}$-labeled PE[P]s (in other words, newly synthesized PE[P]s) was C16:0>C18:1>C22:6>C20:4>C22:4 in control cells (Fig. 5A). This composition differed markedly from the steady-state composition; in particular, the $d_{35}$-labeled PE[P] species contained a higher proportion of C16:0 and a lower proportion of C20:4 than under the steady state (Fig. 4C). In *DHRS7* KO cells, the C18:1 species showed the greatest reduction relative to control cells (average proportion of control level: 36%), followed by C16:0 (64%) and C22:6 (71%) (Fig. 5A). The abundance of the C20:4 species remained largely unchanged, while that of the C22:4 species exhibited a 3.3-fold increase. In contrast, in *DHRS7B* cells, all FA species were reduced to 37–48% of the levels observed in control cells. In DKO cells, the levels ranged from 5% to 20% of those in control cells, depending on the FA species; except for the C22:4 species, the decreases reflected an additive effect of the respective single KOs. The total quantities of $d_{35}$-labeled PE[P]s were 65% and 43% of the control levels in *DHRS7* KO and *DHRS7B* KO cells, respectively (each is the average of two clones), and they were 14% of control levels in *DHRS7/7B* DKO cells (Fig. 5B). In contrast, $d_{35}$-labeled PE levels were almost unchanged in *DHRS7* KO and *DHRS7B* KO cells, and were modestly elevated in *DHRS7/DHRS7B* DKO cells relative to control cells (Fig. 5C). Re-expression of DHRS7 or DHRS7B in the DKO cells restored PE[P] levels close to control levels (Fig. 5D), although expressing these proteins in control cells did not change PE[P] levels. This rescue indicates that the reduced PE[P] abundance in the DKO cells is attributable to the loss of these enzymes. In summary, DHRS7 is involved in plasmalogen biosynthesis in addition to DHRS7B, but its role in the generation of FA species is not identical to that of DHRS7B.

Journal of Cell Science

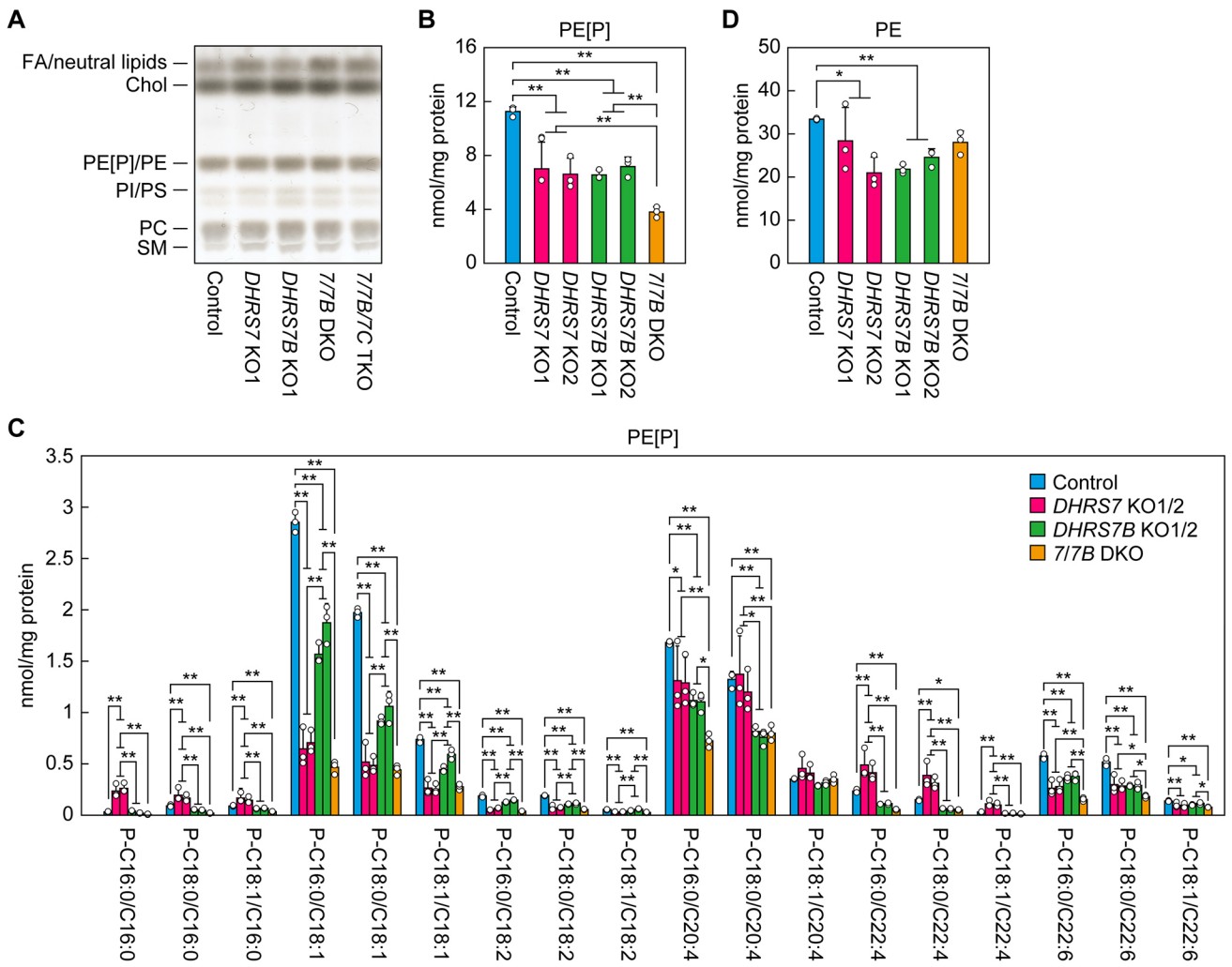

**Fig. 4. Distinct changes in the composition of PE[P] species in *DHRS7* KO and *DHRS7B* KO cells.** (A) Lipids were extracted from the control, *DHRS7* KO (clone 1), *DHRS7B* KO (clone 1), *DHRS7/DHRS7B* DKO and *DHRS7/DHRS7B/DHRS7C* TKO cells, separated via TLC and stained with cupric acetate/ phosphoric acid solution. Chol, cholesterol; PE[P], ethanolamine plasmalogen; PE, phosphatidylethanolamine; PC, phosphatidylcholine; PI, phosphatidylinositol; PS, phosphatidylserine; SM, sphingomyelin. (B-D) Lipids were extracted from the control, *DHRS7* KO (clones 1 and 2), *DHRS7B* KO (clones 1 and 2) and *DHRS7/DHRS7B* DKO cells, and PE[P]s (B,C) and PEs (D) were quantified via LC–MS/MS. Data are mean+s.d. (*n*=3) of the quantities of the indicated species (C) or total quantities (B,D). Statistically significant differences are indicated (Scheffé's test, two-tailed; *P<0.05, **P<0.01). 7/7B DKO, *DHRS7/DHRS7B* DKO; 7/7B/7C TKO, *DHRS7/DHRS7B/DHRS7C* TKO.

## Acyl/alkyl-GnP reductase activity of DHRS7

DHRS7B is known to exhibit activity toward both alkyl-GnP and acyl-GnP (Honsho et al., 2020; James et al., 1997). We next investigated the acyl/alkyl-GnP reductase activity of DHRS7. As the enzyme source, we used total cell lysates from TKO cells overexpressing 3×FLAG-tagged DHRS7 or, for comparison, DHRS7B. Immunoblot analysis revealed that expression levels of 3×FLAG-DHRS7 were 3.1-fold higher than those of 3×FLAG-DHRS7B (Fig. 6A). The alkyl- and acyl-GnP reductase activities were measured by incubating the lysates with hexadecyl-GnP (C16:0 alkyl-GnP) or hexadecanoyl-GnP (C16:0 acyl-GnP) as substrates and NADPH as a co-factor, followed by quantification of the products [1-hexadecyl-LPA (C16:0 1-alkyl-LPA) or 1-hexadecanoyl-LPA (C16:0 1-acyl-LPA), respectively] via LC–MS/MS. Compared with lysates from cells transfected with the empty vector, lysates from cells overexpressing DHRS7 or DHRS7B showed a 7.1-fold or 9.8-fold increase in the quantity of 1-alkyl-LPA, respectively (Fig. 6B). For acyl-GnP reductase activity, lysates from cells overexpressing DHRS7 and DHRS7B produced 61 and 100 times the quantities of

1-acyl-LPA, respectively, of those produced by cells transfected with the empty vector (Fig. 6C). Thus, in our *in vitro* assays, DHRS7B exhibited higher reductase activity than DHRS7, showing 1.4-fold higher alkyl-GnP reductase activity and 1.6-fold higher acyl-GnP reductase activity. Because expression levels of DHRS7 were 3.1-fold higher than those of DHRS7B (Fig. 6A), DHRS7B showed 4.3-fold higher alkyl-GnP reductase activity and 5.0-fold higher acyl-GnP reductase activity than DHRS7 when the enzymatic activities were normalized to their expression levels. Taken together, these findings indicate that DHRS7, like DHRS7B, functions as an acyl/alkyl-GnP reductase capable of acting on both alkyl-GnP and acyl-GnP, and that DHRS7B exhibits stronger enzymatic activity than DHRS7.

## Different intracellular localization of DHRS7 and DHRS7B

To examine the subcellular localization of DHRS7 and DHRS7B, HeLa cells transfected with the plasmid encoding 3×FLAG-tagged *DHRS7* or *DHRS7B* were subjected to indirect immunofluorescence analysis using anti-FLAG antibodies. DHRS7 exhibited a reticular

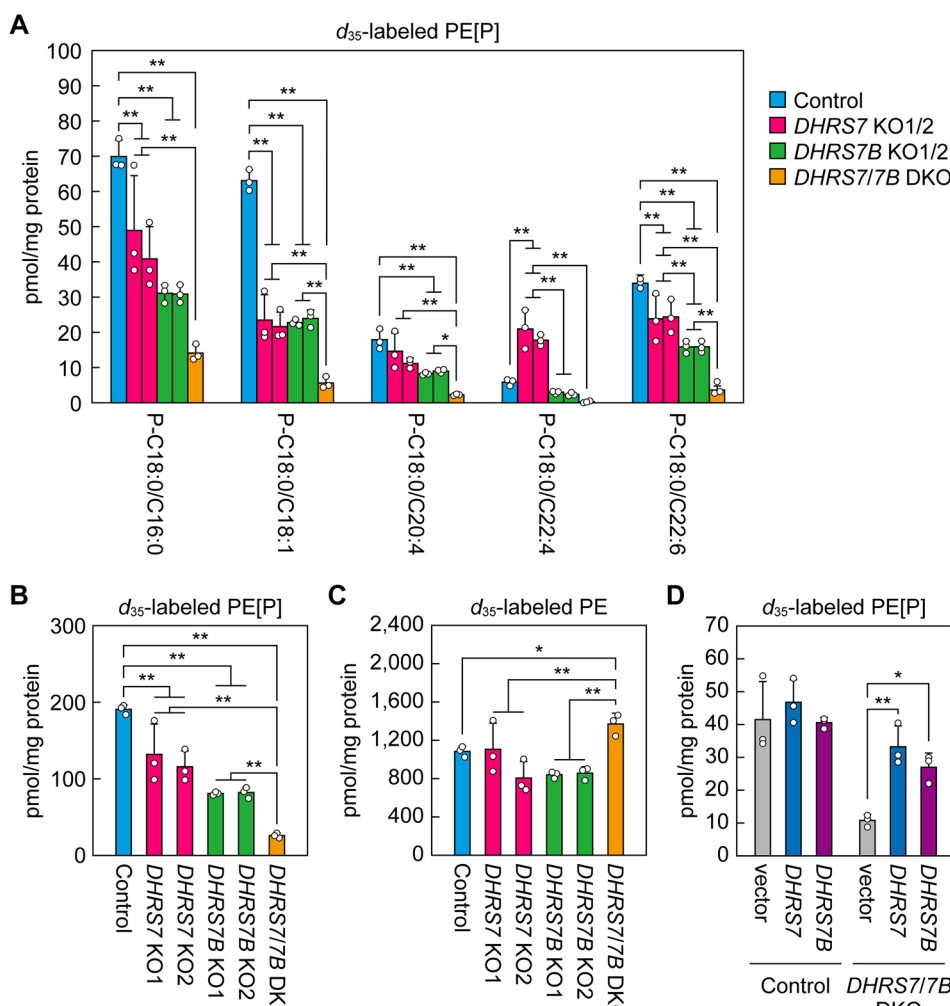

**Fig. 5. Distinct changes in the composition of *de novo* synthesized PE[P] species in *DHRS7* KO and *DHRS7B* KO cells.** (A-C) Control, *DHRS7* KO (clones 1 and 2), *DHRS7B* KO (clones 1 and 2) and *DHRS7/DHRS7B* DKO cells were incubated for 3 h with 50 μM $d_{35}$-stearic acid. Lipids were extracted from the cells and $d_{35}$-labeled PE[P]s (ethanolamine plasmalogens; A,B) and $d_{35}$-labeled PEs (phosphatidylethanolamines; C) were quantified via LC–MS/MS. Data are mean+s.d. ($n$=3) of the quantities of the indicated species (A) or total quantities (B,C). Statistically significant differences are indicated (Scheffé's test, two-tailed; *$P<0.05$, **$P<0.01$). (D) Control cells and *DHRS7/DHRS7B* DKO cells were transfected with an empty vector (pEBMulti-puro 3×FLAG-1) or plasmids encoding *3×FLAG-DHRS7* (pEBMulti-puro 3×FLAG-DHRS7) or *3×FLAG-DHRS7B* (pEBMulti-puro 3×FLAG-DHRS7B). Twenty-four hours after transfection, cells were treated with 2 μg/ml puromycin, and on the following day the medium was replaced with medium containing 1 μg/ml puromycin. One week after transfection, the cells were labeled with 50 μM $d_{35}$-stearic acid for 3 h. Lipids were extracted from the cells and $d_{35}$-labeled PE[P]s were quantified via LC–MS/MS. Data are mean+s.d. ($n$=3) of the total quantities of PE[P]s. Statistically significant differences are indicated (Dunnett's test, two-tailed; *$P<0.05$; **$P<0.01$).

staining pattern characteristic of the ER and indeed colocalized with the ER marker calnexin (Fig. 7). In contrast, DHRS7B showed a punctate intracellular distribution and colocalized with the peroxisomal marker ABCD3. Thus, DHRS7 and DHRS7B showed distinct subcellular localization: DHRS7 in the ER and DHRS7B in peroxisomes.

## DISCUSSION
In this study, we revealed that DHRS7, in addition to the previously known DHRS7B, exhibits acyl/alkyl-GnP reductase activity (Fig. 6) and is involved in plasmalogen biosynthesis (Figs 4 and 5). These proteins showed distinct profiles regarding the PE[P] species involved in synthesis and their intracellular localization. In our results, DHRS7 was localized in the ER, whereas DHRS7B was localized in peroxisomes (Fig. 7). A previous study, however, reported that DHRS7B was primarily localized in peroxisomes, with a small portion localized in the ER (Honsho et al., 2020). The staining pattern of DHRS7B in that study appeared punctate rather than exhibiting the typical reticular structure of the ER, but it is possible that ER-localized DHRS7B resided mainly at peroxisome–ER contact sites. Similarly, the peroxisomal localization of DHRS7B we observed does not preclude its presence at such contact sites. The importance of peroxisome–ER contact sites in plasmalogen biosynthesis has been suggested by the finding that knockdown of any of the ER-localized VAMP-associated proteins

A (VAPA) and B (VAPB) or the peroxisome-localized acyl-CoA-binding domain-containing protein 5 (ACBD5), all of which are involved in contact site formation, reduced plasmalogen production (Hua et al., 2017).

Labeling of *DHRS7* KO cells, *DHRS7B* KO cells and *DHRS7/DHRS7B* DKO cells with $d_{35}$-C18:0 FA revealed that, in the *de novo* PE[P] synthesis pathway, DHRS7 plays a major role in the production of the C18:1 species and contributes to a lesser extent to the synthesis of the C16:0 and C22:6 species (Fig. 5). In contrast, DHRS7B was involved in the production of all PE[P] species examined (C16:0, C18:1, C20:4, C22:4 and C22:6). Because the reaction that introduces a FA into the *sn*-2 position occurs after the ketone group of alkyl-GnPs is reduced to 1-alkyl-LPA by DHRS7 or DHRS7B (Fig. 1), this result was unexpected. The influence of DHRS7 and DHRS7B on PE[P] composition suggests that the 1-alkyl-LPA species produced by these proteins differ in certain respects. In this study, we demonstrate that DHRS7 is localized to the ER, whereas DHRS7B is localized to peroxisomes (Fig. 7), indicating that the site of 1-alkyl-LPA production differs between the two proteins – it occurs in the ER via DHRS7 and in peroxisomes via DHRS7B. This difference in localization is likely to contribute to the resulting distinct profiles of PE[P] species composition. Since DHRS7B exhibits stronger enzymatic activity than DHRS7 (Fig. 6), the production of 1-alkyl-LPA in peroxisomes by DHRS7B may constitute its predominant source, with production in the ER by DHRS7 being a lesser source. Considering that there has

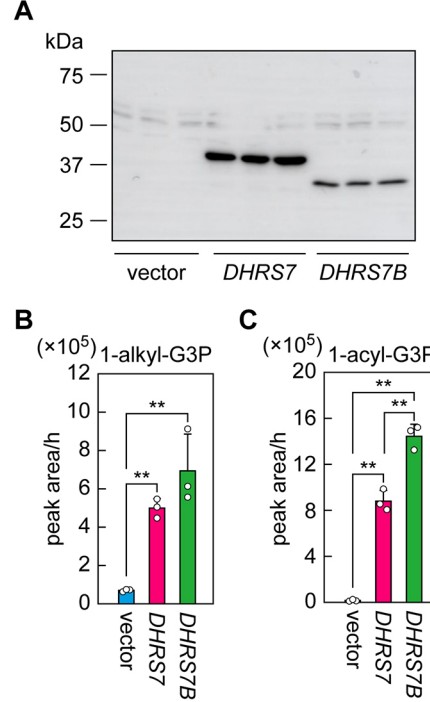

**Fig. 6. Alkyl- and acyl-GnP reductase activities of DHRS7 and DHRS7.** *DHRS7/DHRS7B/DHRS7C* TKO cells were transfected with an empty vector (pEBMulti-puro 3×FLAG-1) or plasmids encoding *3×FLAG-DHRS7* (pEBMulti-puro 3×FLAG-DHRS7) or *3×FLAG-DHRS7B* (pEBMulti-puro 3×FLAG-DHRS7B) and selected with puromycin. (A-C) Total cell lysates were prepared from these cells and subjected to immunoblot analysis using an anti-FLAG antibody (A) or to measurements of alkyl-GnP and acyl-GnP reductase activities (B,C). For the reductase assays, total cell lysates (30 µg protein) were incubated with 25 µM alkyl-GnP (hexadecyl-GnP) (B) or acyl-GnP (hexadecanoyl-GnP) (C) in the presence of 1.25 mM NADPH at 37°C for 2 h. Lipids were extracted and the products (1-hexadecyl-G3P and 1-hexadecanoyl-G3P) were quantified via LC–MS/MS. Data are mean+s.d. (*n*=3). Statistically significant differences are indicated (Tukey's test, two-tailed; **P<0.01).

been no report of lysophospholipid acyltransferase (LPLAT) presence in peroxisomes, it is reasonable to assume that the conversion of 1-alkyl-LPAs to 1-alkyl-PAs occurs in the ER.

Based on these considerations, we propose the following hypothesis to explain why KO of *DHRS7* and *DHRS7B* exerts different effects on the composition of PE[P] species (Fig. 8). The majority of alkyl-GnPs generated in peroxisomes by AGPS are converted to 1-alkyl-LPAs within the same organelle by DHRS7B. 1-Alkyl-LPAs subsequently move to contact sites with the ER, where they are converted to 1-alkyl-PAs by LPLATs. In contrast, alkyl-GnPs that remain unconverted by DHRS7B diffuse from the contact sites throughout the ER membrane (hereafter simply referred to as the ER), where they are metabolized to 1-alkyl-LPAs by DHRS7 and subsequently to 1-alkyl-PAs by LPLATs. Thus, DHRS7 functions as a backup for DHRS7B. In this hypothesis, the site of 1-alkyl-PA production by LPLATs differs, depending on whether the 1-alkyl-LPAs are produced by DHRS7B or DHRS7 – at contact sites in the former case and at the ER in the latter.

We hypothesize that differences in the composition of acyl-CoAs or in the types of LPLATs present at contact sites versus the ER influence the PE[P] species subsequently produced (Fig. 8). Specifically, to explain our results, contact sites either contain acyl-CoAs in greater relative proportions than the ER – C20:4-CoA and C22:4-CoA>C16:0-CoA and C22:6-CoA>C18:1-CoA – or the LPLATs present at contact sites exhibit substrate specificities toward acyl-CoAs in the same order. The acyl-CoA synthetase ACSL4 has been reported to localize not only in the ER but also in peroxisomes (Deng et al., 2025). Because ACSL4 exhibits high levels of activity toward arachidonic acid (C20:4) (Shimbara-Matsubayashi et al., 2019), C20:4-CoA produced by ACSL4 in peroxisomes may migrate to the contact sites with the ER and accumulate there. C22:4-CoA is generated from C20:4-CoA via FA elongation catalyzed by ELOVL2 (Leonard et al., 2002; Ohno et al., 2010). It is possible that a portion of C20:4-CoA present at the contact sites is elongated to C22:4-CoA by ELOVL2 and subsequently utilized for 1-alkyl-PA production there. Although little is known about the LPLATs involved in PE[P] production via the *de novo* pathway, the observation that polyunsaturated FA-containing PE[P] levels were reduced in *LPLAT3* KO cells (Zou et al., 2020) suggests that LPLAT3, at least, is involved in this process (see below for more detail on LPLATs).

According to our hypothesis, the underlying mechanism leading to the characteristic composition of PE[P] species observed in *DHRS7B*

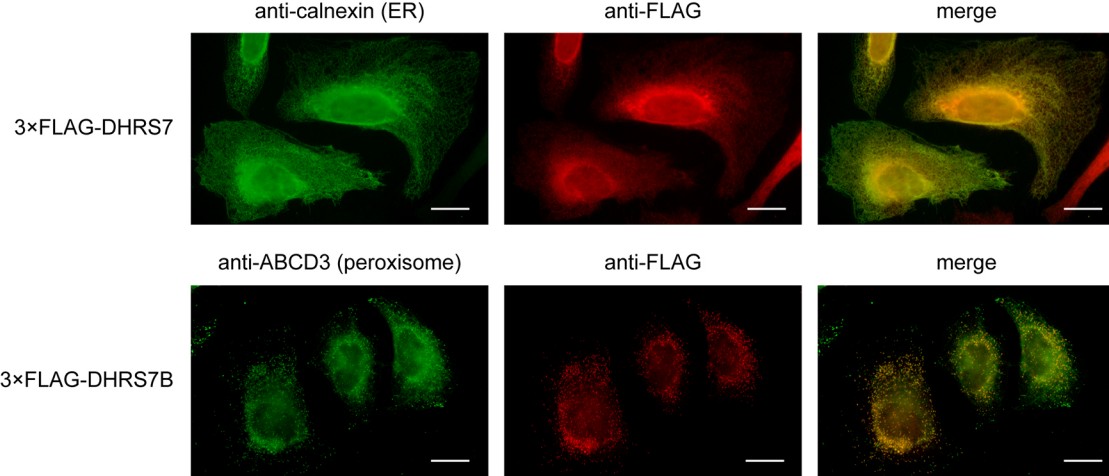

**Fig. 7. Distinct subcellular localization of DHRS7 and DHRS7B.** HeLa cells cultured on coverslips were transfected with plasmids encoding *3×FLAG-DHRS7* (pEFh 3×FLAG-DHRS7) or *3×FLAG-DHRS7B* (pEFh 3×FLAG-DHRS7B). After 24 h, the cells were fixed, permeabilized and immunostained with an anti-calnexin (ER marker), anti-ABCD3 (peroxisome marker) or anti-FLAG antibody, followed by Alexa Fluor-conjugated secondary antibodies. Images were acquired using a fluorescence microscope. Scale bars: 20 µm. The right-most panels show the images from the two panels to the left merged.

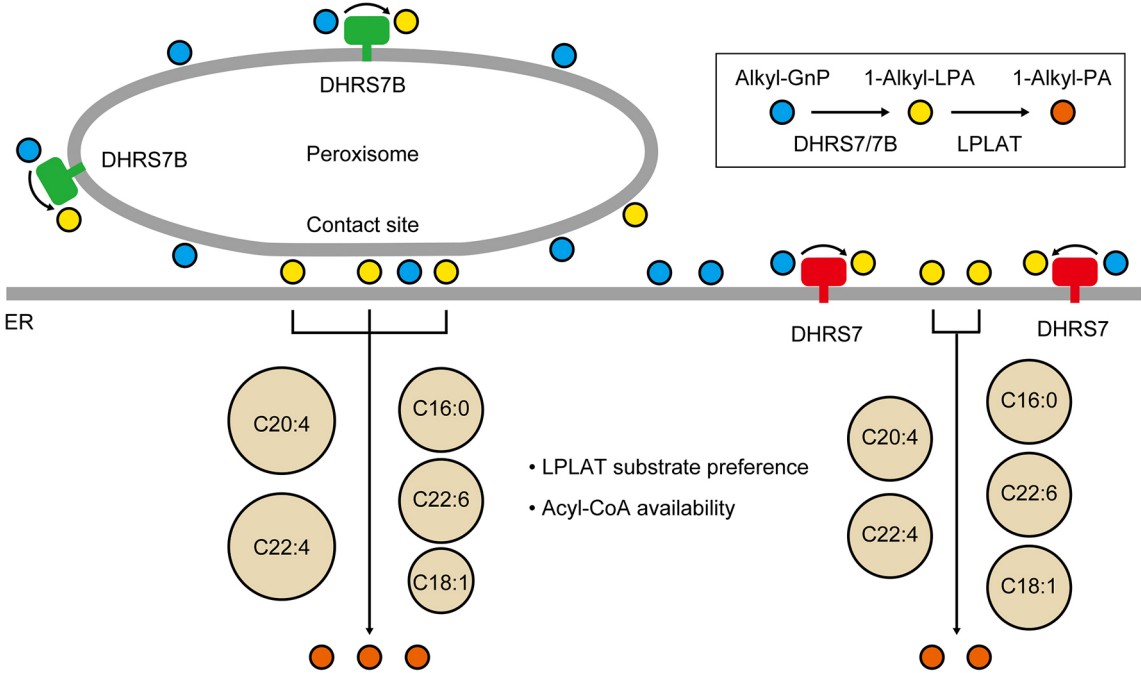

**Fig. 8. Distinct contributions of DHRS7 and DHRS7B to the production of different plasmalogen species.** In peroxisomes, the majority of alkyl-GnPs generated by AGPS are converted to 1-alkyl-LPAs by DHRS7B. 1-Alkyl-LPAs subsequently move to contact sites with the endoplasmic reticulum (ER), where they are converted to 1-alkyl-PAs by LPLATs. In contrast, alkyl-GnPs that remain unconverted diffuse from the contact sites throughout the ER membrane, where they are metabolized to 1-alkyl-LPAs by DHRS7 and subsequently to 1-alkyl-PAs by LPLATs. We hypothesize that differences in the composition of acyl-CoAs or in the types of LPLATs present at contact sites versus the ER influence the PE[P] species subsequently produced.

KO and *DHRS7* KO cells can be explained as follows. In *DHRS7B* KO cells, all alkyl-GnPs are converted into 1-alkyl-LPAs by DHRS7 in the ER, where a wide variety of acyl-CoAs are available or LPLATs with broad substrate specificity are present. However, since DHRS7B exhibits stronger activity than DHRS7 (Fig. 6), the overall alkyl-GnP reductase activity is markedly reduced in *DHRS7B* KO cells, leading to a decrease in the production of 1-alkyl-PAs regardless of the species. In contrast, in *DHRS7* KO cells, all alkyl-GnPs – including those that have previously diffused from the contact site into the entire ER and subsequently returned – are converted into 1-alkyl-LPAs by DHRS7B in peroxisomes. The 1-alkyl-LPAs produced are then transported to the contact sites, where they are converted into 1-alkyl-PAs by LPLATs. However, because the composition of available acyl-CoAs or the substrate specificity of LPLATs differs from that in the ER, the production of certain 1-alkyl-PAs species, such as those containing C18:1 FA, are reduced.

*De novo* synthesized glycerophospholipids, regardless of whether they are ether-type or ester-type, undergo remodeling that alters their *sn*-2 FA composition (Yamashita et al., 1997). Although reports on plasmalogen remodeling are limited, it is known that arachidonic acid is predominantly incorporated during the remodeling process (Ford and Gross, 1994). This is consistent with our present findings. In control cells, the FA composition of newly synthesized ($d_{35}$-labeled) PE[P]s (C16:0, 37%; C18:1, 33%; C20:4, 9%; C22:4, 3%; C22:6, 17%; Fig. 5) differs from that of PE[P]s under steady-state conditions (C16:0, 2%; C18:1, 48%; C20:4, 33%; C22:4, 4%; C22:6, 13%; Fig. 4), indicating that C16:0 species are predominantly remodeled into C20:4 species and, to a lesser, extent, into C18:1 species. In humans, 14 LPLATs have been identified, which include enzymes that exhibit activity toward LPAs or 1-alkyl-LPAs in the *de novo* synthesis pathway (LPA acyltransferase [LPAAT] activity) and those that act on

lysophospholipids containing head groups in the remodeling pathway – for example, LPEAT (lysophosphatidylethanolamine acyltransferase) activity toward lyso-PEs and lyso-PE[P]s (Valentine et al., 2022). Among LPLATs, the primary enzymes exhibiting LPAAT activity are LPLAT1–LPLAT5, which are also known by alternative names such as AGPATs or LPAATs (LPLAT1: AGPAT1 or LPAATα; LPLAT2: AGPAT2 or LPAATβ; LPLAT3: AGPAT3 or LPAATγ1; AGPAT4: LPAATδ; LPLAT5: AGPAT5 or LPAATε) (Valentine et al., 2022). To date, most studies on the LPAAT activity of LPLAT1–LPLAT5 have focused on LPAs, and their activity toward 1-alkyl-LPAs remains largely unknown. Nevertheless, it is likely that LPLAT1–LPLAT5 act redundantly on 1-alkyl-LPAs. As described above, at least LPLAT3 is probably involved in the production of plasmalogens containing polyunsaturated FAs (Zou et al., 2020). Similarly, although knowledge regarding plasmalogen remodeling remains limited, it has been reported that quantities of PE[P] species containing C20:4 and C20:5 are reduced in *Lplat12* (*Lpcat3*) KO mice, suggesting the involvement of LPLAT12 (Thomas et al., 2018).

DHRS7, like DHRS7B, exhibited reductase activity toward not only alkyl-GnP but also acyl-GnP (Fig. 6). 1-Acyl-LPAs produced via the reduction of acyl-GnPs by DHRS7/DHRS7B are utilized for the synthesis of ester-linked glycerophospholipids (Fig. 1). Although the quantity of PEs in *DHRS7* and *DHRS7B* KO cells was lower than in control cells, the extent of this reduction was modest compared with that of PE[P]s, and no reduction was detected in *DHRS7/DHRS7B* DKO cells (Fig. 4). Therefore, the overall contribution of this pathway to ester-linked glycerophospholipid synthesis appears to be low, probably because the pathway that begins with G3P acylation in the ER predominates in 1-acyl-LPA production. Instead, the reductase

Journal of Cell Science

activity of DHRS7/DHRS7B toward acyl-GnP may play an important role in regulating plasmalogen levels. Under physiological conditions, the activity of AGPS, which converts acyl-GnPs to alkyl-GnPs, is likely to be stronger than that of DHRS7/DHRS7B; therefore, acyl-GnPs would probably be converted mainly to alkyl-GnPs and utilized for plasmalogen synthesis. However, if plasmalogen levels increase, the acyl-GnP reductase activity of DHRS7/DHRS7B might also rise, converting acyl-GnPs to 1-acyl-LPAs and thereby suppressing plasmalogen biosynthesis as a possible regulatory mechanism.

In *DHRS7/DHRS7B* DKO cells, the synthesis of PE[P]s was not completely abolished (Figs 4 and 5). The reason for this remains unclear, but it is possible that, in addition to DHRS7 and DHRS7B, there is another as yet unidentified acyl/alkyl-GnP reductase. Although *DHRS7C* was not expressed in the HAP1 cells used in this study (Fig. 3C), and is thus unlikely to contribute here, its protein product can also exhibit acyl/alkyl-GnP reductase activity. Another possibility is the existence of a bypass pathway that does not involve an acyl/alkyl-GnP reductase, as predicted in a study of FAA.K1B cells carrying a *Dhrs7b* mutation (James et al., 1997). For example, there may be a pathway in which alkyl-GnPs are dephosphorylated to form alkyl-glycerones, followed by reduction of the ketone group to produce 1-alkyl-glycerols and subsequently 1-alkyl-2-acyl-glycerols. Indeed, exogenously added radiolabeled 1-alkyl-glycerol has been shown to be metabolized into plasmalogens within cells (Schrakamp et al., 1988). If alkyl-GnPs can be metabolized to 1-alkyl-glycerols via this bypass pathway, plasmalogens could therefore be produced in this way.

Mutations in *GNPAT*, *AGPS*, *FAR1*, the long isoform of *PEX5*, and *PEX7*, which are involved in ether-linked lipid biosynthesis or the transport of proteins to peroxisomes, cause rhizomelic chondrodysplasia punctata (Barøy et al., 2015; Buchert et al., 2014; Purdue et al., 1997; Wanders et al., 1994, 1992). The majority of KO mice for these genes exhibit prenatal or neonatal lethality, although a fraction survive to adulthood (Brites et al., 2011, 2003; Liegel et al., 2014; Pan et al., 2023; Rodemer et al., 2003). Likewise, most *Dhrs7b* KO mice show prenatal lethality, with a small proportion reaching adulthood (Lodhi et al., 2017). These observations suggest that *Dhrs7b* plays a predominant role in ether-linked lipid biosynthesis during embryonic development in mice, whereas *Dhrs7* plays only a minor role. The absence of evidence linking *DHRS7B* to rhizomelic chondrodysplasia punctata might be explained by the greater contribution of *DHRS7* to ether-linked lipid biosynthesis in humans than in mice. However, given their roles in ether-linked lipid biosynthesis, *DHRS7* and *DHRS7B* should be considered in the genetic evaluation of individuals with mild or atypical RCDP-like clinical features.

In summary, we have revealed that DHRS7 is a previously unreported acyl/alkyl-GnP reductase. Future investigations should address the tissue-specific roles of DHRS7 by generating and characterizing of *Dhrs7* KO mice, particularly in tissues where *Dhrs7* expression is high (Fig. 2). In addition, elucidating the molecular mechanisms by which DHRS7 and DHRS7B differentially contribute to the production of distinct PE[P] species and the plasmalogen biosynthetic pathways independent of DHRS7 and DHRS7B will be an important subject for future studies.

## MATERIALS AND METHODS

### Mice
C57BL/6J mice were purchased from Sankyo Labo Service (Sapporo, Japan) and housed under specific pathogen-free conditions at 23±1°C and 50±10% humidity, with a 12 h light and 12 h dark cycle. Water and food (Rodent Diet CE-2; CLEA Japan, Tokyo, Japan) were available *ad libitum*. Animal experiments were approved by the Institutional Animal Care and Use Committee of Hokkaido University (approval number: 22-0036).

### Cell culture and transfection
HAP1 cells (American Type Culture Collection, Manassas, VA, USA) and HeLa cells (RIKEN BioResource Research Center, Tsukuba, Japan) were cultured in Iscove's modified Dulbecco's medium (Thermo Fisher Scientific) and Dulbecco's modified Eagle's medium (D6046; Merck), respectively, both supplemented with 10% fetal bovine serum, 100 U/ml penicillin and 100 µg/ml streptomycin (Merck) at 37°C under 5% $CO_2$. For labeling of *de novo* synthesized PE[P]s in HAP1 cells, 50 µM $d_{35}$-stearic acid (Cayman Chemical) was added to the culture medium. Transfections were performed using Lipofectamine Transfection Reagent with PLUS Reagent (Thermo Fisher Scientific), according to the manufacturer's instructions.

### Quantitative real-time RT-PCR
Expression analyses of *Dhrs7/DHRS7* subfamily members in mouse tissues and HAP1 cells via quantitative real-time RT-PCR were performed as follows. Total RNAs were extracted from tissues of 8-week-old male mice (brain, eyeball, eyelid, tongue, submandibular gland, thymus, lung, heart, stomach, spleen, liver, kidney, small intestine, colon, white adipose tissue, prostate and skeletal muscle) and from HAP1 cells using TRIzol Reagent (Thermo Fisher Scientific) and the NucleoSpin RNA kit (Macherey-Nagel, Düren, Germany), respectively. The RNAs were reverse-transcribed into first-strand cDNAs using the PrimeScript II 1st Strand cDNA Synthesis Kit (Takara Bio, Shiga, Japan) following the manufacturer's instructions. The resulting cDNAs were mixed with gene-specific primer pairs [Table S1; respective forward (-F1) and reverse (-R1) primer pairs] and KOD SYBR qPCR Mix (Toyobo, Osaka, Japan) and subjected to real-time quantitative PCR on the CFX96 Touch Real-Time PCR Detection System (Bio-Rad Laboratories). The mRNA levels in mouse tissues and HAP1 cells were calculated from standard curves generated using serially diluted heart cDNA and HAP1 cDNA, respectively, and normalized with respect to *Hprt1* and *GAPDH*, respectively.

### Generation of *DHRS7* subfamily KO HAP1 cells
*DHRS7* subfamily KO HAP1 cells were generated using the CRISPR-Cas9 system. For constructing the KO vectors, we used pX330A-1×3 (Addgene plasmid #58767), which contains the *Cas9* gene and allows expression of up to three distinct guide RNAs. For generating the *DHRS7* KO plasmid, the guide RNA was designed to target the 20 bases adjacent to the protospacer-adjacent-motif sequence of *DHRS7*. A pair of oligonucleotides (hDHRS7_F2/R2; Table S1) containing the targeted sequence was annealed and cloned into the BbsI site of pX330A-1×3, resulting in the generation of the pTA22 plasmid. The *DHRS7/DHRS7B/DHRS7C* TKO plasmid was constructed as follows. First, pairs of oligonucleotides containing the targeted sequence for *DHRS7B* and *DHRS7C* (hDHRS7B_F2/R2 and hDHRS7C_F2/R2; Table S1) were cloned into the BbsI sites of pX330S-2 and pX330S-3 (Addgene plasmids #58778 and #58779), respectively. Next, DNA fragments containing the guide RNA expression cassettes were isolated from these plasmids via BsaI digestion and cloned in tandem into the BsaI site of pTA22, thereby generating the pTA24 plasmid.

HAP1 cells ($1.0×10^6$ cells per well in a six-well plate) were co-transfected with the pTA24 plasmid and pAK10, which carries a puromycin-resistant gene. After 48 h of incubation, the cells were selected by culturing them in the presence of 2 µg/ml puromycin (Merck) for 24 h. The surviving cells were dissociated from the dishes with 0.25% trypsin/EDTA (Merck) and plated at a density of 10–50 cells per 10 cm dish. After colony formation, individual colonies were isolated, and genomic DNA was extracted from each colony and subjected to PCR with specific primer pairs [Table S1; respective forward (-F3) and reverse (-R3) primer pairs] to amplify the regions containing the guide RNA target sequences. The amplified DNA fragments were analyzed via polyacrylamide gel electrophoresis and Sanger sequencing. From this, *DHRS7/DHRS7B/DHRS7C* TKO cells, *DHRS7/DHRS7B* DKO cells and *DHRS7B* KO cells were established, but *DHRS7* KO cells could not be obtained. Therefore, *DHRS7* KO cells were generated by repeating the experiment using pTA22, which targets only *DHRS7*. Control cells were generated using pX330A-1×3 in an essentially identical manner.

### Cloning of *DHRS7* and *DHRS7B*

The coding sequences of human *DHRS7* and *DHRS7B* were amplified from skeletal muscle cDNA (for *DHRS7*) and testis cDNA (for *DHRS7B*) by PCR using specific primer pairs [Table S1; respective forward (-F4) and reverse (-R4) primer pairs]. The PCR products were cloned into pGEM-T Easy vector (Promega) and their sequences were verified via Sanger sequencing. Each cDNA insert was excised by restriction enzyme digestion, purified via agarose gel electrophoresis and subsequently cloned into two mammalian expression vectors – pEBMulti-puro 3×FLAG-1 (Ohno et al., 2018) and pEFh 3×FLAG-1 (Jojima et al., 2020) – both designed for N-terminal 3×FLAG tagging. The pEBMulti-puro 3×FLAG-1 vector is an episomal vector that replicates in HAP1 cells, enabling prolonged expression of the cloned cDNA, and carries a puromycin-resistant gene for the selection of transfected cells. The resulting plasmids were pEFh 3×FLAG-DHRS7, pEFh 3×FLAG-DHRS7B, pEBMulti-puro 3×FLAG-DHRS7 and pEBMulti-puro 3×FLAG-DHRS7B.

### Immunoblotting

Immunoblotting was performed essentially as described previously (Kitamura et al., 2015). Briefly, total cell lysates were separated via SDS-PAGE and transferred to an Immobilon-P membrane (Merck). The membrane was blocked with Tris-buffered saline [TBS; 20 mM Tris-HCl (pH 7.5) and 137 mM NaCl] containing 5% skim milk and 0.1% Tween 20 (hereafter referred to as blocking solution I) for 1 h at room temperature. The membrane was then incubated with an anti-FLAG polyclonal antibody (generated in-house; no RRID available; 1:1000) (Kitamura et al., 2017) in blocking solution I for 1 h at room temperature. After washing with TBS containing 0.1% Tween 20, the membrane was incubated with a secondary antibody, anti-rabbit IgG, HRP-linked F(ab′)$_2$ fragment (RRID: AB_772191; Cytiva, Little Chalfont, UK; 1:7500), in blocking solution I for 1 h at room temperature. Following washing, labeling was detected using a chemiluminescent reagent prepared by mixing equal volumes of solution A [100 mM Tris-HCl (pH 8.5), 0.4 mM *p*-coumaric acid (Merck), 5 mM luminol (FUJIFILM Wako Pure Chemical)] and solution B [100 mM Tris-HCl (pH 8.5), 0.04% hydrogen peroxide]. Chemiluminescent signals from immunoblots were captured with an Amersham Imager 600 (Cytiva). The full, uncropped immunoblot image is provided in Fig. S2.

### *In vitro* acyl/alkyl-GnP reductase activity assay

For the measurement of acyl/alkyl-GnP reductase activity, *DHRS7/DHRS7B/DHRS7C* TKO cells were transfected with pEBMulti-puro 3×FLAG-DHRS7, pEBMulti-puro 3×FLAG-DHRS7B or empty vector pEBMulti-puro 3×FLAG-1. Twenty-four hours after transfection, the culture medium was replaced with medium supplemented with 1 µg/ml puromycin (Merck) and the cells were maintained for an additional 7 days, with medium changes every 3 days. The cells were scraped in the assay buffer [50 mM sodium phosphate (pH 7.4), 2 mM MgCl$_2$, 1 mg/ml FA-free bovine serum albumin, 1×cOmplete Protease Inhibitor Cocktail (Merck), 1 mM phenylmethylsulfonyl fluoride and 1 mM dithiothreitol] and collected into microtubes. The cells were disrupted by sonication and centrifuged (400 *g*, 3 min, 4°C), and the supernatant was recovered as the total lysate. The reaction mixture consisted of total lysate (30 µg protein), 1.25 mM NADPH and 25 µM hexadecyl-GnP (L-0320; Echelon Biosciences, Salt Lake City, UT, USA) or hexadecanoyl-GnP (L-0310; Echelon Biosciences) in a total volume of 40 µl. The control sample without substrate contained the same volume of ethanol. Samples were incubated at 37°C for 2 h, followed by lipid extraction and quantification of the reaction products (1-hexadecyl-LPA and 1-hexadecanoyl-LPA) via LC–MS/MS, as described below.

### Lipid extraction

Each cell suspension or reaction mixture of *in vitro* acyl/alkyl-GnP reductase activity assay (adjusted to a volume of 100 µl with water) was mixed with 375 µl of chloroform/methanol (1:2, v/v) containing 2.5 pmol of nine-deuterium-labeled PE[P] (P-C16:0/C16:0-$d_9$; Cayman Chemical) as an internal standard. After vigorous mixing, 1 µl of 4 M formic acid was added to neutralize the phosphate group and facilitate efficient extraction of phospholipids into the organic phase, followed by the addition of 125 µl each of chloroform and water. After vigorous mixing, phases were separated

by centrifugation (20,400 *g*, 3 min, room temperature) and the organic phase (lower phase) was recovered and dried. Lipids were dissolved in chloroform/methanol (1:2, v/v) and subjected to LC–MS/MS analysis, as described below. Proteins were recovered from the insoluble intermediate phase and quantified using the Pierce BCA Protein Assay Kit (Thermo Fisher Scientific).

### Lipid analyses

Lipid analysis by TLC was performed using a Silica Gel 60 HPTLC plate (Merck) and a developing solvent consisting of 1-butanol/acetic acid/water (3:1:1, v/v). The plate was sprayed with a copper phosphate reagent [3% (w/v) cupric acetate in 8% (v/v) phosphoric acid], dried and heated at 180°C for 3 min to visualize the lipids.

Lipid analysis by LC–MS/MS was carried out as follows using a Xevo TQ-XS LC-coupled triple quadrupole mass spectrometer (Waters). For the analysis of PE[P]s and PEs, an ACQUITY UPLC CSH C18 reversed-phase column (1.7 µm particle size, 2.1 mm inner diameter, 100 mm length; Waters) was used. LC separation was performed at 0.3 ml/min using a binary gradient of mobile phase A [acetonitrile/water (3:2, v/v) containing 5 mM ammonium formate] and mobile phase B [acetonitrile/2-propanol (1:9, v/v) containing 5 mM ammonium formate], both adjusted to pH 4.0 with formic acid. The gradient steps were as follows: 0 min, 40% B; 0–18 min, linear gradient to 100% B; 18–23 min, 100% B; 23–23.1 min, step to 40% B; and 23.1–25 min, 40% B. For the analysis of 1-hexadecyl-LPA and 1-hexadecanoyl-LPA (products of the acyl/alkyl-GnP reductase assay), LC separation was performed on a YMC-Triart C18 metal-free reversed-phase column (1.9 µm particle size, 2.1 mm inner diameter, 50 mm length; YMC, Kyoto, Japan) at a flow rate of 0.3 ml/min using a binary gradient of mobile phase C (water containing 5 mM ammonium formate) and mobile phase D [water/acetonitrile (1:19, v/v) containing 5 mM ammonium formate], both adjusted to pH 4.0 with formic acid. The gradient steps were as follows: 0–3 min, 45% D; 3–14 min, linear gradient to 100% D; 14–17 min, 100% D; 17–17.1 min, step to 45% D; and 17.1–25 min, 45% D.

All lipids were ionized via electrospray ionization and analyzed in the negative ion mode. MS/MS analysis was performed in the multiple reaction monitoring mode using the *m/z* values of the precursor (Q1) and product (Q3) ions specific to each lipid species, along with optimized cone voltages and collision energies (Tables S2 and S3). Some PE[P] species share identical masses with PE[O] species (e.g. O-C18:1 and P-C18:0), and therefore cannot be distinguished by MS/MS analysis alone. In this study, these species were separated based on their distinct retention times in LC. Data analysis was performed using the MassLynx software (Waters). The quantity of each PE[P] and PE species was calculated from its peak area relative to that of the PE[P] internal standard (P-C16:0/C16:0-$d_9$). The quantity of 1-hexadecyl-LPA or 1-hexadecanoyl-LPA was calculated in the same way using the corresponding external standards (1-hexadecyl-LPA from Echelon Biosciences; 1-hexadecanoyl-LPA from Avanti Research).

### Indirect immunofluorescence microscopy

HeLa cells cultured on coverslips were individually transfected with pEFh 3×FLAG-DHRS7 or pEFh 3×FLAG-DHRS7B. Twenty-four hours after transfection, the cells were fixed with 3.7% formaldehyde in phosphate-buffered saline (PBS) for 10 min at 37°C, followed by permeabilization with 0.1% Triton X-100 in PBS for 5 min at room temperature. The cells were then blocked with PBS containing 10 mg/ml bovine serum albumin (hereafter referred to as blocking solution II) for 30 min at room temperature, followed by incubation with primary antibodies diluted in blocking solution II for 1 h at room temperature. The following primary antibodies were used: anti-FLAG polyclonal antibody (generated in-house; no RRID available; 1:2000) (Kitamura et al., 2017), anti-calnexin 4F10 monoclonal antibody (MBL; RRID: AB_592029; 2.5 µg/ml) and anti-ABCD3 F-1 monoclonal antibody (2.0 µg/ml; Santa Cruz Biotechnology; RRID: AB_2714180; 2.0 µg/ml). After two washes with PBS, the cells were incubated with the following secondary antibodies in appropriate combinations: Alexa Fluor 488-conjugated anti-rabbit IgG antibody (Thermo Fisher Scientific; RRID: AB_2535792; 1:200), Alexa Fluor 594-conjugated anti-rabbit IgG antibody (Thermo Fisher Scientific; RRID: AB_2534095; 1:200) and Alexa Fluor 594-conjugated anti-mouse IgG

antibody (Thermo Fisher Scientific; RRID: AB_2534091; 1:200), each diluted in blocking solution II, for 1 h at room temperature. After two washes with PBS, the cells were mounted on glass slides using ProLong Gold antifade reagent (Thermo Fisher Scientific) and images were acquired using a Leica DM5000B fluorescence microscope.

## Statistical analyses
Statistical analyses were carried using Microsoft Excel and JASP (version 0.95.4; University of Amsterdam, The Netherlands) for Scheffé's test, JASP for Dunnett's test and JMP 13 (SAS Institute,) for Tukey's test. All statistical tests were two-tailed. $P<0.05$ was considered to indicate statistical significance.

## Acknowledgements
The authors used Microsoft M365 Copilot (GPT-5 chat model) in part to improve the readability and language of the article. After using this tool, the authors reviewed and edited the content as needed and take full responsibility for the final version of the manuscript.

## Competing interests
The authors declare no competing or financial interests.

## Author contributions
Conceptualization: T.S.; Funding acquisition: T.S., A.K.; Investigation: T.T., K.O.; Project administration: A.K.; Supervision: A.K., A.K.; Writing – original draft: T.S.; Writing – review & editing: A.K.

## Funding
This work was supported by KAKENHI Grants-in-Aid for Scientific Research (JP22H04986 to A.K. and JP23K24020 to T.S.) from the Japan Society for the Promotion of Science. Open Access funding provided by Hokkaido University. Deposited in PMC for immediate release.

## Data and resource availability
All materials generated in this study are available from the corresponding author upon reasonable request. All relevant data and details of resources are provided within the article and its supplementary information.

## Peer review history
The peer review history is available online at https://journals.biologists.com/jcs/lookup/doi/10.1242/jcs.264759.reviewer-comments.pdf

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
