## [Peer Review File · Journal of Cell Science]

The acyl- and alkyl-glycerone phosphate reductase reductase DHR57 is involved in the production of distinct plasmalogen species from DHR57B

Tenga Takahashi, Kento Otsuka, Takayuki Sassa and Akio Kihara

DOI: 10.1242/jcs.264759

Editor: Robert Parton

Review timeline

Original submission:	29 January 2026
Editorial decision:	24 February 2026
First revision received:	9 March 2026
Accepted:	27 March 2026

Original submission

First decision letter

MS ID#: jcs.264759

MS TITLE: Alkylglycerone phosphate reductase DHR57 is involved in the production of different plasmalogen species from DHR57B

AUTHORS: Tenga Takahashi; Kento Otsuka; Takayuki Sassa; Akio Kihara

ARTICLE TYPE: Research Article

Dear Dr Kihara,

We have now reached a decision on the above manuscript.

As you will see, the reviewers raise a number of substantial criticisms that prevent me from accepting the paper at this stage. This includes addressing a number of crucial issues including the discrimination of plasmenyl vs plasmanyl ether species. They suggest, however, that a revised version might prove acceptable, if you can address their concerns. If you think that you can deal satisfactorily with the criticisms on revision, I would be pleased to see a revised manuscript. We would then return it to the reviewers.

Reviewer 1: Plasmalogens are an understudied, structurally unique group of ether lipids that constitute mammalian membranes and are hypothesized to play important roles in membrane dynamics and cell signaling. Characterizing the enzymes involved in the pathway for mammalian ether lipid and plasmalogen synthesis is key to uncovering their regulation, subcellular distribution, and metabolism; in particular, understanding the distribution of polyunsaturated species is crucial to determining what role these lipids play in regulating membrane dynamics. This study contributes to the field's understanding of this role by characterizing a novel alkyl/acylglycerone phosphate reductase, DHR57, that produces distinct plasmalogen species and has a different subcellular localization than the already-characterized enzyme DHR57B. This study isolates the effects of three distinct alkyl/acylglycerone phosphate reductases involved in the ether lipid biosynthetic pathway, DHR57, DHR57B, and DHR57C, by developing KO HAP1 cells via CRISPR-Cas9 genome editing and using TLC and LC-MS/MS lipidomics methods with deuterated fatty acid labeling to quantify changes in their lipid composition. Their findings of reduced, but not eliminated, labeled plasmalogen species in the KO cells and significantly diminished levels of labeled plasmalogens with 18:1, 16:0, and 22:4 sn-2 tails in the DHR57 KO cells suggests that there are multiple pathways towards

plasmalogen biosynthesis and that DHR57 has a specificity towards producing those particular species. The study also uses fluorescence microscopy to localize tagged DHR57B to the ER and tagged DHR57B to the peroxisome, showing differences in subcellular localization of these enzymes and suggesting the potential for regulation of plasmalogen trafficking via their activity, to be investigated in future studies.

Overall, this is a strong, clearly written, and straightforward study that provides an important contribution to the field of ether lipid biosynthesis and metabolism. There is strong evidence characterizing the activity, subcellular localization, and tissue distribution of the DHR57 enzyme subfamily, as well as changes in overall plasmalogen species distribution following disruptions in the biosynthetic pathway. The study claims to have characterized distinct plasmalogen species via LC-MS/MS lipidomics; however, the methods do not detail efforts to distinguish plasmalogen (P-PE) from alkyl ether (O-PE) species of PE, which have identical fragmentation patterns, so it is possible that the reported plasmalogen species do not represent the true measured plasmalogen distribution of the cells. Additionally, the study reports tissue distribution of the DHR57 subfamily of enzymes via RT-PCR of mouse tissue that does not align with previously reported tissue distribution of plasmalogens, suggesting the need for further investigation of this difference.

Comments:

1. Tissue distribution of DHR57 subfamily in mice

In the literature, plasmalogen distribution has been shown to be localized strongly in the brain and heart in humans and in the brain, colon, heart, and testes in mice (Braverman and Moser, BBA, 2012; Koch et al. Anal. Chem., 2020). However, the tissue distribution reported in this study via analysis of mRNA isolated from mice via RT-PCR finds almost no DHR57 mRNA in the brain and heart, with the highest levels found in eyelid, submandibular gland, prostate, skeletal muscle, and white adipose tissue. If the DHR57 subfamily is crucial to plasmalogen synthesis, as this study suggests, but these enzymes have little activity in plasmalogen-enriched tissues, then perhaps their activity is tissue-specific and limited to organs/tissues with low plasmalogen levels. The authors should explain this discrepancy, or provide additional data on plasmalogen levels in the mouse tissues alongside DHR57 subfamily mRNA levels.

2. Lipidomics: plasmalogen standard

In the "Lipid analyses" portion of the Methods section, the study reports quantifying PE[P] and PE species by calculating peak areas relative to that of a PE[P] internal standard (P-C16:0/C16:0-d9). The authors should report where this internal standard was supplied from.

3. Lipidomics: Distinguishing O-PE and P-PE species

Discrimination between plasmanyl (alkyl ether, O-PX) and plasmeryl (vinyl ether, P-PX) ether species is not possible by exact masses and fragmentation patterns alone using high mass resolution LC-MS/MS methods due to their identical fragmentation spectra (Koch et al., see Figure 3, Table S3, Table S4, Figure S2). Therefore, it is essential to distinguish these lipids based on their retention times via reverse-phase liquid chromatography. This study reports m/z values of precursor (Q1) and product (Q3) ions but does not validate whether these peaks correspond to plasmalogens alone or rather represent isobaric plasmalogen and alkyl ether species. The authors should report retention times for each species identified as a plasmalogen and show an example chromatogram from which fragmentation patterns were characterized, from both samples and their reported internal P-PE standard, such that it can be validated that they are not reporting alkyl ether lipids as plasmalogens.

Reviewer 2: In this manuscript, the authors investigate whether DHR57, a member of the DHR57 subfamily of short-chain dehydrogenases/reductases, contributes to ether lipid biosynthesis alongside the established alkyl/acyl-glycerone phosphate reductase DHR57B. The study identifies DHR57 as a novel contributor to ether lipid biosynthesis and proposes a model in which spatial compartmentalisation (ER versus peroxisomes/contact sites) contributes to species-specific plasmalogen profiles. Overall, the manuscript is mechanistically interesting and potentially suitable for publication after revision. However, several conceptual and technical issues should be addressed to strengthen the conclusions.

Major comments

1. The entire functional analysis is performed in HAP1 cells. Given that plasmalogens are particularly enriched in neural and cardiac tissues, the use of a hematopoietic-derived cell line warrants clearer justification. The authors should: i) provide explicit rationale for selecting HAP1 cells to study ether lipid biology, ii) indicate whether HAP1 cells have been previously used for studies of ether lipid biosynthesis, iii) discuss the limitations of this model system in the Discussion section.
2. In the *in vitro* reductase assays (Fig. 6), the authors conclude that DHRS7B likely has stronger enzymatic activity than DHRS7. However, DHRS7B protein expression appears lower than DHRS7 in the lysates. To support the claim that DHRS7B has stronger intrinsic activity, enzymatic activity should be normalised to protein expression levels (e.g., by densitometric quantification of the FLAG signal). Alternatively, the statement should be softened to avoid overinterpretation.
3. The central mechanistic conclusion is that DHRS7 preferentially contributes to C18:1-containing PE[P] species in the *de novo* pathway. However, the reductase step precedes acylation at the sn-2 position, and fatty acid incorporation depends on LPLAT activity and the availability of acyl-CoA pools. Thus, the observed species differences likely reflect spatial metabolic coupling rather than intrinsic substrate selectivity of DHRS7 for specific alkyl-GnP species. Although the authors partially acknowledge this point, the manuscript would benefit from more clearly distinguishing between "species preference" and "spatial metabolic routing," rephrasing statements to avoid implying direct fatty acid selectivity at the reductase level, and emphasising that the observed differences likely arise downstream of 1-alkyl-LPA generation.
4. The localisation data show ER-like reticular staining and colocalisation with calnexin. However, no quantitative colocalisation analysis is presented, and there is no evidence excluding partial peroxisomal localisation. Given that peroxisome-ER contact sites are central to the proposed model, higher-resolution imaging and/or quantitative colocalisation analysis would strengthen this conclusion.
5. The causal relationship between gene knockout and the lipid phenotype would be strengthened by re-expression (rescue) of DHRS7 in DHRS7 KO cells and demonstration that PE[P] species are restored. Although enzymatic overexpression experiments are shown, functional lipid rescue is not demonstrated. This represents a significant gap and should be addressed experimentally if feasible.
6. The authors report that both PE[P] and PE levels are reduced in single KO cells, whereas PE levels are largely unchanged in DKO cells. This raises an important mechanistic question: why are ester-type PEs reduced in DHRS7 or DHRS7B single KOs if the primary defect lies in alkyl/acyl-GnP reduction? The manuscript would benefit from a clearer mechanistic explanation for why both PE[P] and PE are reduced in single KOs but not in DKO cells. At minimum, this observation should be discussed.
7. The manuscript does not clearly describe how PE[P] species were distinguished from plasmalogen (PE[O]) species and diacyl PEs in the LC-MS/MS workflow. Was a published protocol followed? If so, it should be cited. If not, additional methodological detail should be provided to ensure reproducibility and clarity regarding plasmalogen identification.

Minor comments

1. In the Introduction, ether-linked glycolipids are described as including seminolipids and GPI anchors. However, ether-linked triacylglycerols (TG[O] species) have also been reported. The authors should clarify or refine this statement.
2. Throughout the manuscript, the term "plasmalogen synthesis" is used in contexts that more broadly concern ether lipid or ether phospholipid synthesis. The terminology should be adjusted to avoid conceptual narrowing of the pathway.
3. Gene names (e.g., *Dhrs7*, *Dhrs7b*) should be italicised when referring to genes (particularly in mouse). Please ensure that nomenclature conforms to journal standards.

4. The manuscript alternates between "alkyl-GnP reductase" and "alkyl/acyl-GnP reductase." The terminology should be clarified early and used consistently throughout.
5. The manuscript briefly discusses RCDP. Given the clinical importance of ether lipid deficiencies, the translational implications of DHR57 should be more explicitly addressed.
6. The manuscript is generally well written; however, portions of the Results and Discussion sections are lengthy and could be streamlined for clarity and focus.

First revision

Author response to reviewers' comments

Dear Dr. Parton and reviewers,

Thank you very much for reviewing our manuscript (manuscript number: jcs.264759) and providing useful comments. We have revised the text and figures accordingly and below provide itemized responses to all comments.

We are grateful to the reviewers for recognizing the significance of our study and for providing these constructive comments. We believe that the manuscript has been significantly improved by their input and hope that it is now acceptable for publication in the *Journal of Cell Science*.

Reviewer 1

Overall Comment: *“Plasmalogens are an understudied, structurally unique group of ether lipids that constitute mammalian membranes and are hypothesized to play important roles in membrane dynamics and cell signaling. Characterizing the enzymes involved in the pathway for mammalian ether lipid and plasmalogen synthesis is key to uncovering their regulation, subcellular distribution, and metabolism; in particular, understanding the distribution of polyunsaturated species is crucial to determining what role these lipids play in regulating membrane dynamics. This study contributes to the field's understanding of this role by characterizing a novel alkyl/acylglycerone phosphate reductase, DHR57, that produces distinct plasmalogen species and has a different subcellular localization than the already-characterized enzyme DHR57B. This study isolates the effects of three distinct alkyl/acylglycerone phosphate reductases involved in the ether lipid biosynthetic pathway, DHR57, DHR57B, and DHR57C, by developing KO HAP1 cells via CRISPR-Cas9 genome editing and using TLC and LC-MS/MS lipidomics methods with deuterated fatty acid labeling to quantify changes in their lipid composition. Their findings of reduced, but not eliminated, labeled plasmalogen species in the KO cells and significantly diminished levels of labeled plasmalogens with 18:1, 16:0, and 22:4 sn-2 tails in the DHR57 KO cells suggests that there are multiple pathways towards plasmalogen biosynthesis and that DHR57 has a specificity towards producing those particular species. The study also uses fluorescence microscopy to localize tagged DHR57B to the ER and tagged DHR57C to the peroxisome, showing differences in subcellular localization of these enzymes and suggesting the potential for regulation of plasmalogen trafficking via their activity, to be investigated in future studies.*

Overall, this is a strong, clearly written, and straightforward study that provides an important contribution to the field of ether lipid biosynthesis and metabolism. There is strong evidence characterizing the activity, subcellular localization, and tissue distribution of the DHR57 enzyme subfamily, as well as changes in overall plasmalogen species distribution following disruptions in the biosynthetic pathway. The study claims to have characterized distinct plasmalogen species via LC-MS/MS lipidomics; however, the methods do not detail efforts to distinguish plasmalogen (P-PE) from alkyl ether (O-PE) species of PE, which have identical fragmentation patterns, so it is possible that the reported plasmalogen species do not represent the true measured plasmalogen distribution of the cells. Additionally, the study reports tissue distribution of the DHR57 subfamily of enzymes via RT-PCR of mouse tissue that does not align with previously reported tissue distribution of plasmalogens, suggesting the need for further investigation of this difference.”

Response

Thank you very much for your positive assessment. We have provided detailed responses to the reviewer's concerns regarding the discrimination between P-PE and O-PE species and the tissue distribution of the *Dhrs7* subfamily, as outlined below.

Comment 1: “Tissue distribution of DHR57 subfamily in mice. In the literature, plasmalogen distribution has been shown to be localized strongly in the brain and heart in humans and in the brain, colon, heart, and testes in mice (Braverman and Moser, *BBA*, 2012; Koch et al. *Anal. Chem.*, 2020). However, the tissue distribution reported in this study via analysis of mRNA isolated from mice via RT-PCR finds almost no DHR57 mRNA in the brain and heart, with the highest levels found in eyelid, submandibular gland, prostate, skeletal muscle, and white adipose tissue. If the DHR57 subfamily is crucial to plasmalogen synthesis, as this study suggests, but these enzymes have little activity in plasmalogen-enriched tissues, then perhaps their activity is tissue-specific and limited to organs/tissues with low plasmalogen levels. The authors should explain this discrepancy, or provide additional data on plasmalogen levels in the mouse tissues alongside DHR57 subfamily mRNA levels.”

Response

We consider that DHR57 proteins are not responsible for the rate-limiting step of plasmalogen biosynthesis; therefore, their expression levels do not necessarily correlate with plasmalogen abundance. In fact, overexpression of DHR57 or DHR57B in cells did not increase plasmalogen levels (new Fig. 5D), suggesting that once the amount of DHR57/DHR57B exceeds a certain threshold, it no longer affects plasmalogen levels.

In Fig. 2, we compared the expression levels relative to the housekeeping gene *Hprt1*. Generally, expression levels greater than 10% of a housekeeping gene are considered relatively high to high, 1-10% are regarded as low to moderate, and less than 1% are considered very low to low. In our study, the expression levels of *Dhrs7b* in the brain, heart, and colon were 12.7%, 25.3%, and 82.2% of *Hprt1* expression, respectively, corresponding to relatively high to high expression.

What is important in Fig. 2, in our view, is the difference in expression among members of the *Dhrs7* subfamily within each tissue. To present these differences more accurately and prevent tissues with moderate expression from appearing nearly unexpressed when plotted together with tissues having extremely high expression, we reorganized Fig. 2 into three graphs with appropriately adjusted y-axis scales. We have also updated the manuscript text in the *Results* section to clearly convey that relative expression within each tissue is the key point of Fig. 2.

Comment 2: “Lipidomics: plasmalogen standard. In the “Lipid analyses” portion of the *Methods* section, the study reports quantifying PE[P] and PE species by calculating peak areas relative to that of a PE[P] internal standard (P-C16:0/C16:0-d9). The authors should report where this internal standard was supplied from.”

Response

We have already described in the *Lipid extraction* section of the original manuscript that we used “nine-deuterium-labeled PE[P] (P-C16:0/C16:0-d₉; Cayman Chemical, Ann Arbor, MI, USA)” as the internal standard.

Comment 3: “Lipidomics: Distinguishing O-PE and P-PE species. Discrimination between plasmalogen (alkyl ether, O-PX) and plasmalogen (vinyl ether, P-PX) ether species is not possible by exact masses and fragmentation patterns alone using high mass resolution LC-MS/MS methods due to their identical fragmentation spectra (Koch et al., see Figure 3, Table S3, Table S4, Figure S2). Therefore, it is essential to distinguish these lipids based on their retention times via reverse-phase liquid chromatography. This study reports m/z values of precursor (Q1) and product (Q3) ions but does not validate whether these peaks correspond to plasmalogen alone or rather represent isobaric plasmalogen and alkyl ether species. The authors should report retention times for each species identified as a plasmalogen and show an example chromatogram from which fragmentation patterns were characterized, from both samples and their reported internal P-PE standard, such that it can be validated that they are not reporting alkyl ether lipids as plasmalogens.”

Response

As the reviewer pointed out, some PE[P] species share identical masses with PE[O] species (for example, O-C18:1 and P-C18:0), making them indistinguishable by MS/MS alone. However, PE[P]s and PE[O]s exhibit clearly different retention times in LC because they differ in both the type of double bond (*trans* in PE[P]s vs. *cis* in PE[O]s) and the position of the double bond (C-1 for PE[P]s vs. n-9/6/3 for PE[O]s). In this study, we distinguished PE[P]s from PE[O]s based on these differences in retention time. We have added this information to the LC-MS/MS section of the Materials and Methods in the revised manuscript.

Reviewer 2

Overall Comment: *“In this manuscript, the authors investigate whether DHRS7, a member of the DHRS7 subfamily of short-chain dehydrogenases/reductases, contributes to ether lipid biosynthesis alongside the established alkyl/acyl-glycerone phosphate reductase DHRS7B. The study identifies DHRS7 as a novel contributor to ether lipid biosynthesis and proposes a model in which spatial compartmentalisation (ER versus peroxisomes/contact sites) contributes to species-specific plasmalogen profiles. Overall, the manuscript is mechanistically interesting and potentially suitable for publication after revision. However, several conceptual and technical issues should be addressed to strengthen the conclusions.”*

Response

Thank you very much for your positive evaluation. We have addressed the conceptual and technical concerns raised by the reviewer, and our responses and revisions are provided below.

Major Comment 1: *“The entire functional analysis is performed in HAP1 cells. Given that plasmalogens are particularly enriched in neural and cardiac tissues, the use of a hematopoietic-derived cell line warrants clearer justification. The authors should: i) provide explicit rationale for selecting HAP1 cells to study ether lipid biology, ii) indicate whether HAP1 cells have been previously used for studies of ether lipid biosynthesis, iii) discuss the limitations of this model system in the Discussion section.”*

Response

HAP1 cells were chosen primarily because their haploid genome makes gene knockout experiments highly efficient. For our study, which required generating multiple *DHRS7*-subfamily knockout lines, this technical advantage was essential. Importantly, plasmalogens are present in virtually all mammalian cells, although their abundance varies across tissues. Therefore, while neural and cardiac cells are enriched in plasmalogens, investigating the biosynthetic pathway itself does not necessarily require such cells. Our aim was to dissect the enzymatic steps in ether lipid synthesis, rather than to study tissue-specific physiological functions of plasmalogens.

As for the prior study using HAP1 cells, HAP1 cells have been previously used to study plasmalogen biosynthesis, including in the characterization of TMEM189, the enzyme responsible for converting plasmanyl lipids into plasmalogens (doi:10.1073/pnas.1917461117). This supports the suitability of HAP1 cells as a model system for investigating plasmalogen biosynthetic enzymes.

Major Comment 2: *“In the in vitro reductase assays (Fig. 6), the authors conclude that DHRS7B likely has stronger enzymatic activity than DHRS7. However, DHRS7B protein expression appears lower than DHRS7 in the lysates. To support the claim that DHRS7B has stronger intrinsic activity, enzymatic activity should be normalised to protein expression levels (e.g., by densitometric quantification of the FLAG signal). Alternatively, the statement should be softened to avoid overinterpretation.”*

Response

In our *in vitro* assays, DHRS7B exhibited higher reductase activity than DHRS7 (1.4-fold higher alkyl-GnP reductase activity and 1.6-fold higher acyl-GnP reductase activity). The protein amount of DHRS7 was 3.1-fold higher than that of DHRS7B in the lysates. When these enzymatic activities were normalized to the relative protein expression levels, DHRS7B showed 4.3-fold higher alkyl-GnP reductase activity and 5.0-fold higher acyl-GnP reductase activity compared with DHRS7. These quantitative values have been added to the *Results* section.

Major Comment 3: *“The central mechanistic conclusion is that DHRS7 preferentially contributes to C18:1-containing PE[P] species in the de novo pathway. However, the reductase step precedes acylation at the sn-2 position, and fatty acid incorporation depends on LPLAT activity and the*

availability of acyl-CoA pools. Thus, the observed species differences likely reflect spatial metabolic coupling rather than intrinsic substrate selectivity of DHR57 for specific alkyl-GnP species. Although the authors partially acknowledge this point, the manuscript would benefit from more clearly distinguishing between "species preference" and "spatial metabolic routing," rephrasing statements to avoid implying direct fatty acid selectivity at the reductase level, and emphasising that the observed differences likely arise downstream of 1-alkyl-LPA generation."

Response

Our manuscript does not propose intrinsic substrate selectivity of DHR57 for specific alkyl-GnP species. From the beginning, our interpretation has been that the observed differences in PE[P] species reflect spatial metabolic routing arising from the distinct subcellular localization of DHR57 and DHR57B, rather than substrate preference at the reductase step. Accordingly, none of our statements imply fatty acid selectivity by the reductase; instead, they consistently attribute the differences to downstream processes following 1-alkyl-LPA formation.

Major Comment 4: *"The localisation data show ER-like reticular staining and colocalisation with calnexin. However, no quantitative colocalisation analysis is presented, and there is no evidence excluding partial peroxisomal localisation. Given that peroxisome-ER contact sites are central to the proposed model, higher-resolution imaging and/or quantitative colocalisation analysis would strengthen this conclusion."*

Response

Our imaging was performed using standard fluorescence microscopy rather than high resolution systems, which are not available in our laboratory. Nevertheless, the ER-like reticular pattern and clear overlap with calnexin are sufficient to establish that DHR57 is predominantly localized to the ER. Even if a minor fraction of DHR57 were present on peroxisomes, this would not alter our interpretation, because our model does not require exclusive ER localization but relies on the predominant distribution and spatial organization relative to DHR57B. We also note that the reviewer does not provide a rationale for why quantitative colocalization analysis is necessary in this context; for determining predominant ER localization, qualitative fluorescence imaging is generally adequate. Therefore, we consider the current imaging data sufficient to support our conclusion.

Major Comment 5: *"The causal relationship between gene knockout and the lipid phenotype would be strengthened by re-expression (rescue) of DHR57 in DHR57 KO cells and demonstration that PE[P] species are restored. Although enzymatic overexpression experiments are shown, functional lipid rescue is not demonstrated. This represents a significant gap and should be addressed experimentally if feasible."*

Response

We have already generated rescue data in which either *DHR57* or *DHR57B* was re-expressed in the *DHR57/7B* DKO cells. In both cases, re-expression restored the PE[P] levels, demonstrating that the lipid phenotype is indeed caused by loss of these enzymes. Although the reviewer specifically requested rescue in the *DHR57* single KO background, we do not have corresponding data for that line. However, because the DKO cells exhibit a more pronounced phenotype and rescue by either enzyme fully restores PE[P] levels, we consider the DKO rescue experiment an appropriate and informative alternative that addresses the causal relationship between gene loss and lipid phenotype. These data have been added as Fig. 5D.

Major Comment 6: *"The authors report that both PE[P] and PE levels are reduced in single KO cells, whereas PE levels are largely unchanged in DKO cells. This raises an important mechanistic question: why are ester-type PEs reduced in DHR57 or DHR57B single KOs if the primary defect lies in alkyl/acyl-GnP reduction? The manuscript would benefit from a clearer mechanistic explanation for why both PE[P] and PE are reduced in single KOs but not in DKO cells. At minimum, this observation should be discussed."*

Response

The PE levels in the single KO and DKO cells do not differ significantly. Although PE levels in the single KOs are significantly lower than in the control, the absence of a significant difference between the control and the DKO does not imply that the single KO and DKO groups differ from

each other. These comparisons involve independent statistical tests, and a significant difference in one comparison and a lack of significance in another cannot be used to infer a difference between the non-compared groups. Consistent with this, the PE level in *DHRS7* KO1 is essentially the same as that in the DKO line. Moreover, the effect of *DHRS7* or *DHRS7B* deletion on PE levels is limited compared with the much stronger and more consistent reduction observed in PE[P] species. The modest reduction in PE seen in the single KOs is likely due to indirect influences, such as compensatory changes in gene expression or alterations in lipid homeostasis. We have added this interpretation to the revised manuscript.

Major Comment 7: *“The manuscript does not clearly describe how PE[P] species were distinguished from plasmalogen (PE[O]) species and diacyl PEs in the LC-MS/MS workflow. Was a published protocol followed? If so, it should be cited. If not, additional methodological detail should be provided to ensure reproducibility and clarity regarding plasmalogen identification.”*

Response

As explained in our response to Reviewer 1 (Comment 3), some PE[P] species share identical masses with PE[O] species, making them indistinguishable by MS/MS alone. However, PE[P] species and PE[O] species exhibit clearly different retention times in LC because they differ in the type and position of their double bond(s). In this study, we distinguished PE[P] species from PE[O] species based on these retention-time differences. We have added this information to the LC-MS/MS section of the Materials and Methods in the revised manuscript.

Minor Comment 1: *“In the Introduction, ether-linked glycolipids are described as including seminolipids and GPI anchors. However, ether-linked triacylglycerols (TG[O] species) have also been reported. The authors should clarify or refine this statement.”*

Response

We have revised the Introduction to include ether-linked triacylglycerols (TG[O] species).

Minor Comment 2: *“Throughout the manuscript, the term “plasmalogen synthesis” is used in contexts that more broadly concern ether lipid or ether phospholipid synthesis. The terminology should be adjusted to avoid conceptual narrowing of the pathway.”*

Response

We have carefully reviewed the terminology used throughout the manuscript. We ensured that the terms “plasmalogen,” “ether glycerolipid,” and “ether glycerophospholipid” are used in a context-dependent manner, selecting the term that most accurately reflects the specific lipid class being discussed.

Minor Comment 3: *“Gene names (e.g., *Dhrs7*, *Dhrs7b*) should be italicised when referring to genes (particularly in mouse). Please ensure that nomenclature conforms to journal standards.”*

Response

We have thoroughly reviewed the entire manuscript to ensure that gene names are presented in accordance with standard nomenclature. Throughout the text, gene symbols have been italicized when referring to genes, and we have applied the appropriate species-specific conventions: human gene names are written in all uppercase letters (e.g., *DHRS7*), whereas mouse gene names begin with an initial capital letter followed by lowercase letters (e.g., *Dhrs7*). Corrections have been made wherever necessary.

Minor Comment 4: *“The manuscript alternates between “alkyl-GnP reductase” and “acyl/alkyl-GnP reductase.” The terminology should be clarified early and used consistently throughout.”*

Response

We have carefully reviewed the terminology used throughout the manuscript. The term “acyl/alkyl-GnP reductase” is now used consistently when referring to the enzyme in a general sense as well as when describing reductase activity toward both acyl-GnP and alkyl-GnP. In contrast, when referring specifically to activity toward a single substrate class, we use “acyl-GnP reductase” or “alkyl-GnP reductase” as appropriate. The manuscript has been revised to ensure consistent, context-dependent usage of these terms.

Minor Comment 5: “*The manuscript briefly discusses RCDP. Given the clinical importance of ether lipid deficiencies, the translational implications of DHR57 should be more explicitly addressed.*”

Response

Although the translational implications of DHR57 are difficult to define with certainty, we have added the following sentence to the *Discussion*:

“However, given their roles in ether-linked lipid biosynthesis, *DHR57* and *DHR57B* should be considered in the genetic evaluation of patients with mild or atypical RCDP-like clinical features.”

Minor Comment 6: “*The manuscript is generally well written; however, portions of the Results and Discussion sections are lengthy and could be streamlined for clarity and focus.*”

Response

We reviewed the requested *Results* and *Discussion* sections and simplified portions of the Results. However, we were unable to find parts of the *Discussion* that could be removed without weakening the arguments we aimed to convey. The original manuscript contained 5,083 words, which is well within the journal’s 8,000-word limit; therefore, we considered reduction unnecessary.

Second decision letter

MS ID#: jcs.264759R1

MS Title: Acyl/alkyl-glycerone phosphate reductase DHR57 is involved in the production of distinct plasmalogen species from DHR57B

Authors: Tenga Takahashi; Kento Otsuka; Takayuki Sassa; Akio Kihara

Article Type: Research Article

Dear Dr Kihara,

I am happy to tell you that your manuscript has been accepted for publication in Journal of Cell Science, pending standard publication integrity checks.